

# Changes in precipitation and temperature patterns related to the state of the North Atlantic Ocean during the Medieval Climate Anomaly

Shailendra Pratap[1], Yannis Markonis[1], and Cécile L. Blanchet[2]

[1]Faculty of Environmental Sciences, Czech University of Life Sciences Prague, Kamýcká 129, Praha– Suchdol, 165 00, Czech Republic
[2]Section Geomorphology, Helmholtz Centre Potsdam, GFZ German Research Centre for Geosciences, Potsdam, Germany

**Correspondence:** Shailendra Pratap (pratap@fzp.czu.cz)

**Abstract.**

In a warmer climate, uncertainties persist regarding regional precipitation responses and a potential weakening of the Atlantic Meridional Overturning Circulation (AMOC). This study examines the Medieval Climate Anomaly (~800–1399 CE) warm period to uncover hydroclimate patterns and their links with the North Atlantic Ocean variability, including AMOC, Sea Surface Temperature (SST), and the Inter-Tropical Convergence Zone (ITCZ) at centennial (100-years) scales. Analyzing change-sensitive multi-proxy data reveals that North Atlantic Ocean conditions play a significant role in influencing hydroclimate variability across Europe and North America, potentially by regulating atmospheric heat and moisture transport. Specifically, we show that warm SST conditions correspond to warmer climates on both continents, while low SST periods are associated with a southward shift of the ITCZ, potentially initiating cooler climates and hydrological variations. However, the state of the AMOC remains unclear, despite indications of subtle weakening in some records. Exploring hydroclimate suggests that continental-scale precipitation variations are linked to temperature changes, but regional responses are uncertain. Notably, warmer/slightly warmer climates are primarily linked to more humid conditions, especially in mid-latitude regions. Conversely, slightly colder climates tend to result in more arid conditions. Comparing model assimilation with proxy data reveals significant discrepancies, suggesting that either the models fail to adequately capture key processes or the proxy data contain substantial uncertainties. While our findings provide valuable insights into regional hydroclimate changes and variability in the North Atlantic Ocean state under a warmer climate, they also emphasize the necessity for more in-depth research on regional precipitation variability and the identification of appropriate proxies for tracking AMOC signals.

## 1 Introduction

A warmer climate will have far-reaching impacts on socioeconomic life by altering the hydrological cycle and affecting water resources. In recent few decades, significant changes have occurred in the global hydrological cycle (Hobeichi et al., 2022), and it is expected that future global warming will cause even more alterations in precipitation patterns (Thackeray et al., 2022). Most studies project an intensification of the hydrological cycle in warmer climates (Trenberth et al., 2003; Douville et al.,





2021). This is because warm conditions increase the atmospheric moisture holding limit, which will lead to more intense precipitation (Held and Soden, 2006). However, some studies have also documented drier conditions in a warmer climate (Dai, 2013; Huang et al., 2016). Owing to these conflicting findings, the behavior of the hydrological cycle in a warm climate remains enigmatic, although there is substantial evidence of its intensification as temperature increases in the paleoclimatic records (Pratap and Markonis, 2022; Konecky et al., 2023).

In addition to directly modulating the hydrological cycle, warmer conditions also exert strong indirect controls over oceanic states. Specifically, a warmer climate influences glaciers and Sea Surface Temperature (SST), leading to the melting of ice sheets and sea ice, respectively. This melting subsequently affects global ocean circulation patterns, thereby impacting hydroclimate variables such as temperature and precipitation patterns (Zhuravleva et al., 2023). The discharge of freshwater into the North Atlantic Ocean from melting ice masses is known to disturb ocean circulation patterns (Kuhlbrodt et al., 2007), leading to cascading impacts on both regional and global hydroclimate systems. In a warmer climate, the changes observed in SST and the Atlantic Meridional Overturning Circulation (AMOC) compared to the preceding decades have drawn significant global attention. The thermohaline circulation, i.e., the AMOC is a chain of ocean currents system that carries warm and salty surface-waters northwards (from the Equatorial regions towards the Arctic). These surface waters subsequently sink down and lose buoyancy over the Arctic region and move southwards (from the Arctic towards Equatorial regions) as cold deep-waters. During this circulation process, the AMOC redistributes atmospheric heat and moisture, which exerts a strong control on the global water cycle (Marshall et al., 2014; Latif et al., 2022) and influencing hydroclimate distribution. Recently, the state of the AMOC has drawn wide attention because research indicates that it is slowing down, and numerical models project a further weakening by 2100 (Boers, 2021; Masson-Delmotte et al., 2021).

There is an ongoing debate on the prevailing hypothesis that increasing global warming will weaken the AMOC (Weijer et al., 2020; Boers, 2021). Due to the increasing rate of greenhouse gases and the accompanying warm climate (Latif et al., 2022), freshwater discharge into the ocean alters the density of the surface water, making it more buoyant and less likely to sink. Consequently, a decrease in the North Atlantic surface water density will likely inhibit the ocean deep-water convection and will cause a weakening of the AMOC (Böning et al., 2016). Rahmstorf (1999) and Caesar et al. (2018) suggested that the AMOC is presently in its weakest state during the past millennium, due to increasing anthropogenic global warming (Bakker et al., 2016). While a shutdown of AMOC has been projected before the end of the present century (Boers, 2021), the possibility remains uncertain and still largely debated (Kriegler et al., 2009).

Importantly, the variability of the AMOC has a significant impact on hydroclimates in North American and European regions (Jackson et al., 2015). The anticipated weakening of the AMOC is likely to result in reduced atmospheric moisture transport towards Europe and North America, leading to limited precipitation and cold conditions in these regions (Liu et al., 2020; Vellinga and Wood, 2002). Moreover, a slowdown of the AMOC could impact the monsoon system over Asia and Africa (Gupta et al., 2003), as well as alter moisture flows and air temperatures globally by pushing away Inter-Tropical Convergence Zone (ITCZ) southward (Couchoud et al., 2009; Brovkin et al., 2021). A significant weakening of the AMOC and the consequent southward displacement of the ITCZ would likely lead to global hydroclimatic alterations, such as shifts in hydrological cycle patterns across the terrestrial regions (Liu et al., 2020).



The evidence from the distant past reveals that most of the abrupt events were in response to the Atlantic long-term variability, particularly the AMOC in specific (Weijer et al., 2019). For instance, the events such as Dansgaard-Oeschger (Santos et al., 2020), the last ice age, and the deglaciation period were all responses to alterations in the AMOC (Lynch-Stieglitz, 2017). A weakened AMOC has been hypothesized in the Heinrich events (Allen et al., 1999), the Younger Dryas (Clark et al., 2012), the 8.2 ka events (Ellison et al., 2006), and particularly cold climatic phases (Aguiar et al., 2021). The majority of studies conducted thus far have focused on cooler time intervals. Interestingly, AMOC shifts often occurred during the transition time between glacial-interglacial (Bellomo et al., 2021) or interglacial-glacial periods.

It is still being investigated whether the current warmer climate will lead to colder climates in the future, and what relationship exists between warmer climates and AMOC variability. Therefore, understanding the relationship between AMOC variability and hydrological cycle variability, including SST and ITCZ response, in warm climates could help predict current and future hydroclimate shifts. Since SST exert a substantial influence on the distribution of hydroclimate patterns across the North Atlantic regions, a comprehensive examination of SST patterns in conjunction with hydroclimate patterns may yield valuable insights into its significance (Sutton and Hodson, 2005).

However, only a few observational studies have focused on the behavior of AMOC, SST, and ITCZ, including their relationships with hydroclimate patterns during past warm climates, mainly the Medieval Climate Anomaly (MCA) period. The lack of data on past AMOC variability poses a challenge for examining AMOC variations and their possible influences on hydroclimate conditions. Additionally, climate models do not fully capture past climatic conditions and ocean circulation, which hinders their ability to provide reliable conclusions and projections (Srokosz and Bryden, 2015). Consequently, AMOC variability during the MCA is still not well understood. Nevertheless, investigating the relationship between past warm climates, AMOC, and SST could be a foundation for understanding how these parameters will influence the global hydroclimate in a warm climate. Therefore, our study stands as unique due to our focus on investigating North Atlantic variability (including SST, AMOC, and ITCZ shift) and its correlation with terrestrial hydroclimate patterns during the MCA period, which was a more or less warm period like the ongoing interglacial time.

Since the core period of the MCA falls within 1000 and 1300 CE, we selected a time scale from 800 to 1399 CE. This broader temporal scope enables us to examine variations occurring not only during the MCA but also in the periods preceding and succeeding this distinct climatic phase. The climate during the MCA is often suggested to have predominantly warm and arid conditions, specifically during the core period, and cold and humid conditions afterward (LaMarche Jr, 1974; Graham et al., 2011). Despite the widespread acceptance of these hydroclimatic changes, the factor responsible for the abrupt decline in average temperature during the MCA-Little Ice Age (LIA) transition remains elusive. However, it has been suggested that shifting global atmospheric circulation patterns between the MCA and LIA were the main drivers of hydroclimatic changes (Lamb, 1969; Gonzalez-Rouco et al., 2011), which still needs to be tested. Our study aims to investigate changes in precipitation, mainly precipitation minus evaporation (P-E), and temperature patterns in response to centennial-scale variations in the state of the North Atlantic, using regional multi-proxies.





## 2 Data and Methods

### 2.1 Data

For this study, we used a variety of proxy reconstructions available in open-source databases, specifically the NOAA's National Centers for Environmental Information (https://www.ncei.noaa.gov/access/paleo-search/) and the Past Global Changes (PAGES; https://pastglobalchanges.org/science/data/databases). We chose precipitation and temperature-sensitive material, including stalagmites, lake sediments, tree rings, ice cores, marine sediments, and peatland sediments, which are distributed across North America and Europe. These materials serve as indirect indicators of changes in P-E patterns over the locations where the materials are collected. So when we refer to precipitation, it is actually denoting P-E state. Additional information about the selected material and proxies with their specific locations can be found in Figure 1. To ensure consistent and accurate representation of the relevant variables, such as temperature and precipitation (i.e., P-E), we maintained alignment with the specific variables as indicated by the original investigators in their respective data sources. Further elaboration and details on datasets can be found in the corresponding sections, namely Table A1, Table A2, Table A3, and Table A4.

For choosing AMOC-sensitive proxies, we selected a domain over the North Atlantic regions, defined by latitude -12.9 to 87°E and longitude 5 to -90°N. Since direct measurements of AMOC started in 2004 (McCarthy et al., 2015), and instrumental data are not available for the period we investigated, we utilized well established AMOC-sensitive tracers. These included deep-sea proxies such as $\delta^{13}$C, cadmium/calcium ($Cd/Ca$) ratios, and sortable silts (mean size). We have chosen to focus on these tracers for a specific reason– they serve as paleomicronutrient indicators, crucial for identifying patterns related to the migration and dynamics of deep-water masses, as these nutrients are transported via oceanic circulation (Marchitto Jr et al., 2002). Changes in the concentration of these nutrients serve as markers of the state of oceanic circulation. Monitoring these tracer concentrations in deep seawater is essential for understanding the dynamics of ocean circulation and for assessing the stability of the global ocean circulation system over time. Notably, the observed reduction in $\delta^{13}$C values and $Cd/Ca$ ratio within deep/intermediate seawater serves as a significant indicator of the decreased presence of North Atlantic Deep Water (Bell et al., 2014; Valley et al., 2022). Furthermore, sortable silts ($10\text{-}63\mu m$) are associated with the physical strength of deepwater flow, as they are transported by ocean currents/water masses and changes over time. Consequently, these sortable silts are valuable for providing an overview of deep ocean currents and serve as a valuable proxy for the strength of the AMOC (Hoffmann et al., 2019).

In addition, we also examined the SST and ITCZ variations to determine if they provide signals of the AMOC variation. To track the ITCZ, we reconstructed the ITCZ shift index using the $\delta^{18}$O values recorded from stalagmite by various investigators. For this reconstruction, we used 11 $\delta^{18}$O records from different sites located in the Northern Hemisphere (NH) and 5 $\delta^{18}$O records from various sites in the Southern Hemisphere (SH) due to data availability limitations. All selected sites are currently positioned within the present migration path of the ITCZ across both the NH and SH. After collecting these records, the mean $\delta^{18}$O value was calculated separately for the NH and SH. Then, the SH mean was subtracted from the NH mean to obtain the ITCZ shift index. For comparison, the ITCZ shift index reconstructions by Tan et al. (2019) and Chawchai et al. (2021) were also utilized. Changes in SST may be linked to variability in sea-ice cover, the reorganization of ocean surface currents,



freshwater input by melting of ice-sheets and intricate feedback processes between the atmosphere, land, and ocean. Moreover, the findings regarding ITCZ and SST variability will serve as a crucial references for anticipating future changes in the North Atlantic and their correlated impacts on the hydroclimate. In tracking SST conditions, magnesium-calcium ($Mg/Ca$) ratios and $U^{k'}37$ proxies are widely recognized for their effectiveness (Bendle and Rosell-Melé, 2004; Wei et al., 2000). Hence, we have incorporated these proxies into our study to assess variations in SST. Additionally, data assimilation based reconstruction

on ITCZ and SST from Steiger et al. (2018) have been included to assess the reliability of model-based paleoclimate outputs.

For the comparative analysis between proxies and models, assimilation data related to paleoclimate were acquired from two sources: the Paleo Hydrodynamics Data Assimilation product (PHYDA; for temperature, SST, ITCZ, and Palmer Drought Severity Index (PDSI) as an indicator of humid and arid conditions) (Steiger et al., 2018) and PaleoView (for temperature and precipitation) (Fordham et al., 2017). The PDSI is widely used to determine soil moisture availability, with higher values sug-

135 gesting increased precipitation/wet conditions, whereas lower PDSI values denote low precipitation/dry conditions. Therefore, in this study, the PDSI is utilized as an indicator of humid and arid conditions. PHYDA is a global hydroclimate reconstruction generated by combining about 2978 proxy time series with the physical constraints of atmosphere-ocean climate models. The PHYDA provides data from 1 to 2000 CE with annual means (April to March), the boreal growing season of June, July, and August (JJA), and the austral growing season of December, January, and February (DJF). In contrast, PaleoView uses

the TRaCE21ka experiment based on the daily simulated outcomes of the Community Climate System Model ver.3 (CCM3) (Collins et al., 2006). It provides monthly and annual scale temperature and precipitation data from 22 ka BP to 1989 CE. We gained the climate model data at the annual scale, covering the time scale of 800-1399 CE, for the specified variables.





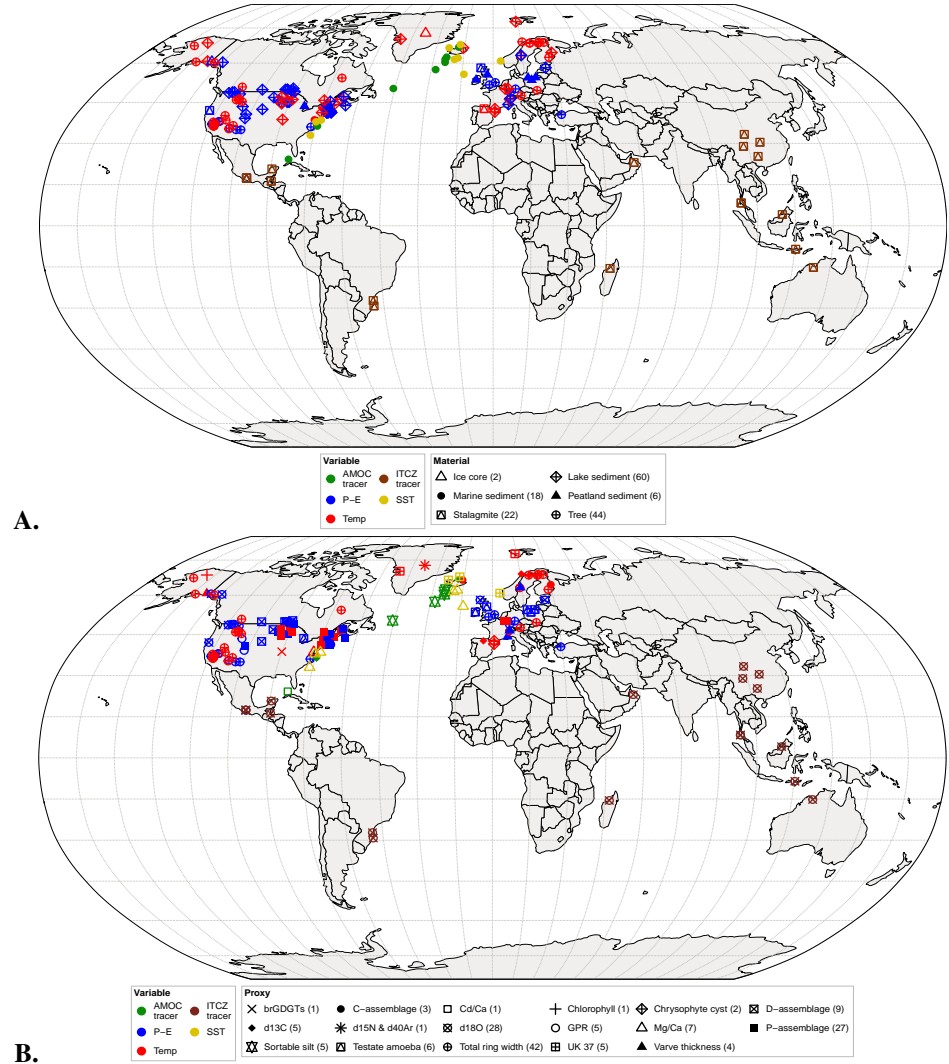

**Figure 1.** The map provided in this figure displays the spatial distribution of data sources categorized as material sources ('A') and proxy sources ('B'). These sources were utilized in our study to collect information related to precipitation minus evaporation (P-E), temperature (Temp), sea surface temperature (SST), Atlantic Meridional Overturning Circulation (AMOC), and the records of the Inter-Tropical Convergence Zone (ITCZ). In plot B, 'D-assemblage' denotes diatom assemblage data, 'P-assemblage' represents pollen data, and 'C-assemblage' corresponds to chironomid data. The numbers in the parentheses indicate the quantity of proxies.

## 2.2 Methods

After the data mining, we implemented measures to ensure the datasets were consistent. In our study, we utilized diverse
datasets, each with distinct units of measurement. To establish consistency and express all the data on a uniform scale, we applied a natural log transformation. Furthermore, we identified negative values in datasets, and to address this, we converted





all values to positive using Equation 1 and subsequently performed a log transformation on them.

$$y(x) = X + |\min(x)| + 1 \tag{1}$$

In Equation 1, $y(x)$ denotes the transformation that is employed to ensure that the data possesses positive values suitable for a log transformation. $|\min(x)|$ represents the minimum value within the dataset, as an absolute value. Finally, $X$ represents the values within the data that are being converted into positive values through this transformation.

The values of $\delta^{18}O$ allow us to assess past hydrological conditions, distinguishing between humid and arid periods and reflecting the prevailing precipitation conditions. Therefore, we ensured that the $\delta^{18}O$ records were accurately represented in our analysis. Several studies have suggested a negative correlation between isotopic values (from $\delta^{13}C$ and $\delta^{18}O$) and precipitation conditions (Hong et al., 2001; Tan et al., 2019), indicating that low isotopic values correspond to increased precipitation conditions and vice versa. To align the values with this relationship, we applied a negative log transformation to reverse the $\delta^{13}C$ and $\delta^{18}O$ records, ensuring that low isotopic values correspond to low precipitation and high isotopic values correspond to high precipitation. After ensuring the correct transformation of values in all records, we standardized them by converting them into z-scores. Subsequently, these standardized values were aggregated into centennial time steps for each dataset individually. We then employed these standardized deviation values to estimate variability on a continental scale.

Finally, we performed a spatiotemporal investigation of precipitation and temperature, conducting time-series analysis (including model assimilated records, AMOC, SST, and ITCZ) to assess overall continental scale variability. To this end, we employed maps of Europe and North America to visually illustrate the regional distribution of variables, facilitating our understanding of their relationships and interconnections. The spatiotemporal representation of data proved invaluable for examining latitudinal and spatial shifts in hydroclimate regimes. All analyses were conducted using the R statistical software (R Core Team et al., 2022).

## 3 Results

### 3.1 Variability in AMOC, SST, and ITCZ during the MCA warm period

Our analysis of AMOC-sensitive tracers, at the centennial scale, indicates notably weak and/or non-significant changes in AMOC strength (Moffa-Sánchez et al., 2019). We employed multiple records to track AMOC variations, each presenting distinct signals of AMOC changes over time. The potential weakening of AMOC due to warmer climates remains uncertain (Kilbourne et al., 2022), given the divergence in responses observed across various records. It is possible that either the changes are very low or very short, or the proxies are not sensitive enough to capture AMOC weakening responses. In our dataset, specifically, both $\delta^{13}C$ and $Cd/Ca$ exhibit weak signals of AMOC after the mid-$10^{th}$ century (Figure 2-A and B). Among the sortable silt records (Figure 2 C), three (NEAP-4k, Orphan Knoll, GS06-14408GC) indicate weak signals of AMOC from the $10^{th}$ to $11^{th}$ century, while two (MD99-2251 and ODP983) present a contrasting response during this period. We observed that sortable silt records from MD99-2251 and ODP983 revealed an early indication of a weak AMOC signal compared to $\delta^{13}C$





and $Cd/Ca$. Specifically, this signal emerges from the $9^{th}$ century and extends from the mid-$9^{th}$ century to approximately the mid-$10^{th}$ century, respectively.

We observed that SST (Figure 3-A) over the North Atlantic region was high from the $9^{th}$ to the mid-$10^{th}$ centuries. However, thereafter, SST started to decline, and between the mid-$10^{th}$ and $14^{th}$ centuries, the average SST remained low. Notably, the SST conditions remained relatively stable and consistent with the centennial mean from the late $12^{th}$ to mid-$13^{th}$ century. Subsequently, from the mid-$13^{th}$ century onward, there was a noticeable weakening observed in SST conditions. Furthermore, the confidence interval (CI) and kernel density (KD) estimation suggest statistically significant variation in SST between the

$9^{th}$ to $14^{th}$ centuries (Figure 3-A). Upon analyzing the significance of SST records using the CI, it becomes evident that except for the $10^{th}$ and $11^{th}$ century, there is high uncertainty in the SST variability during the $9^{th}$ and from the $12^{th}$ to $14^{th}$ centuries. Additionally, the KD estimates for the records from the $9^{th}$ and $10^{th}$ centuries indicate that the SST condition was higher than their centennial mean, as indicated by the $Mg/Ca$, $U^{k'}37$, and PHYDA records. However, a few $Mg/Ca$ and $U^{k'}37$ records show a below-average response. Later, during the $11^{th}$ to $14^{th}$ century, the lowered values of $Mg/Ca$, $U^{k'}37$, and PHYDA

indicate low SST conditions, while a few $Mg/Ca$, $U^{k'}37$ records suggest the opposite.

In the case of the ITCZ, we observed that its variability was synchronous with fluctuations in SST. Our findings indicate that southward/northward shifts in the ITCZ are significantly associated with weak/strong phases of the AMOC. Notably, while the PHYDA ITCZ assimilation exhibits fluctuations similar to those of the proxy data, it demonstrates greater variability than the proxy observations.





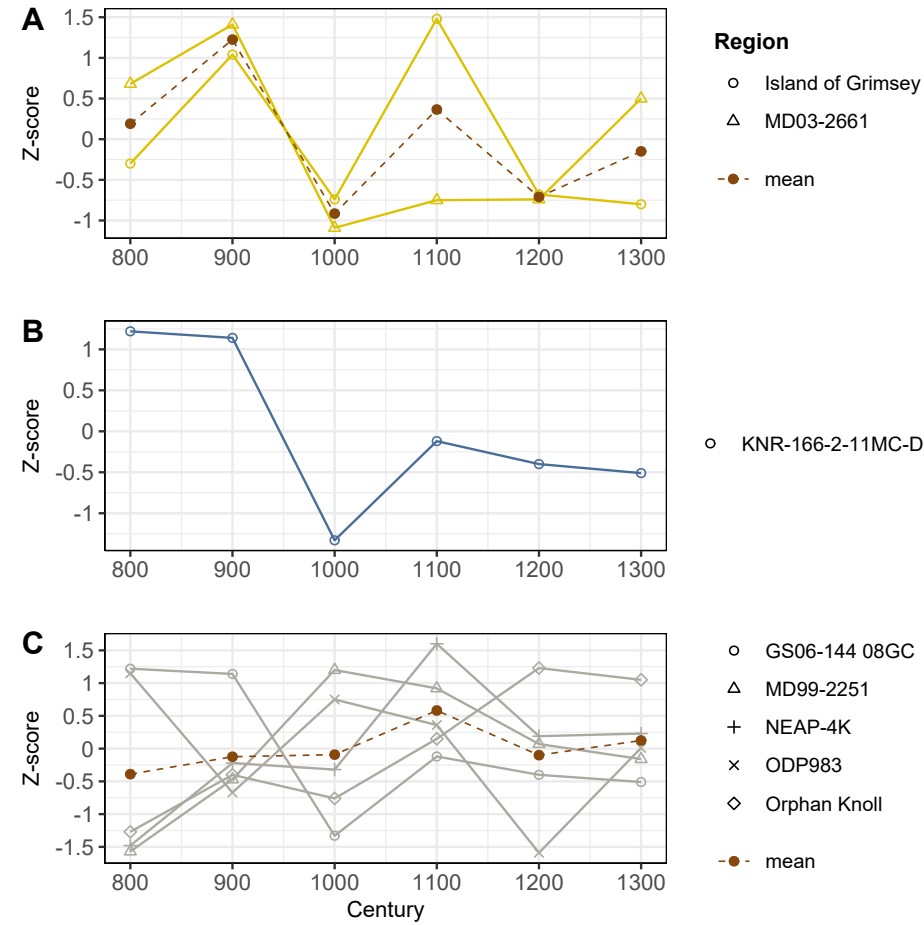

**Figure 2.** State of AMOC in the North Atlantic regions. Variations in $\delta^{13}$C (A), $Cd/Ca$ ratio (B), and sortable silt (C). The mean reflects the comprehensive average of all records.





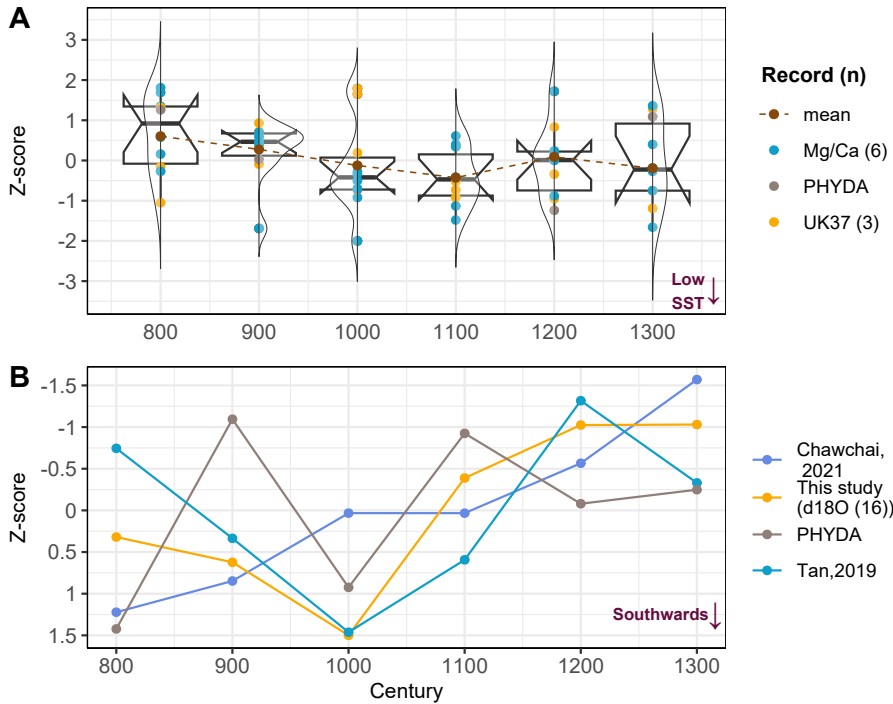

**Figure 3.** SST (A) and ITCZ (B) conditions over North Atlantic. Positive values for ITCZ indicate a southward shift and negative values suggest a northward shift. The 'Record (n)' denotes the proxy and the number of proxies employed for the analysis.

## 3.2 Mean temporal patterns of hydroclimates

While observing hydroclimate conditions across Europe at a continental scale, we show that temperature records reveal warm conditions until the $11^{th}$ century, followed by cold conditions until the $14^{th}$ century (Figure 4-A). Conversely, precipitation patterns at the continental scale suggest predominantly wet conditions until the $10^{th}$ century (Figure 4-B). After this period, although variations in precipitation persist, they tend to fluctuate around their centennial mean values, consistently remaining below that centennial mean. The continental scale analysis suggests a correlation between warm climates and wet conditions, while cold climates correspond to dry conditions (Figure 4) . However, at the regional level (Figure 6), hydrological conditions appear to be more closely influenced by variability in ocean-atmospheric circulation, local thermodynamic changes, and the availability of water sources (Chen et al., 2019; Pratap and Markonis, 2022).

Our analysis of the CI distribution (Figure 4-A and B) highlights the complexity and heterogeneity of climatic responses. For temperature (Figure 4-A), the analysis reveals notable variations across the centuries. In the $10^{th}$ century, there is a warming trend, but the change is not statistically significant as the confidence intervals (CIs) overlap with those of adjacent centuries. Conversely, the mid-$11^{th}$ century exhibits a pronounced shift towards colder values, with the CI not overlapping with those of preceding centuries, indicating a significant cooling event during this period. In the case of P-E conditions (Figure 4-B), the CIs indicate relatively stable conditions from the $10^{th}$ to the $13^{th}$ century. However, significant changes occur in the $9^{th}$





and $12^{th}$ centuries. The $9^{th}$ century shows a shift towards drier conditions, with the CI not overlapping with those of adjacent centuries, highlighting a significant drying trend. The $12^{th}$ century, however, shows a shift towards wetter conditions, with the CIs slightly overlapping with those of the adjacent centuries, suggesting this change is not statistically significant.

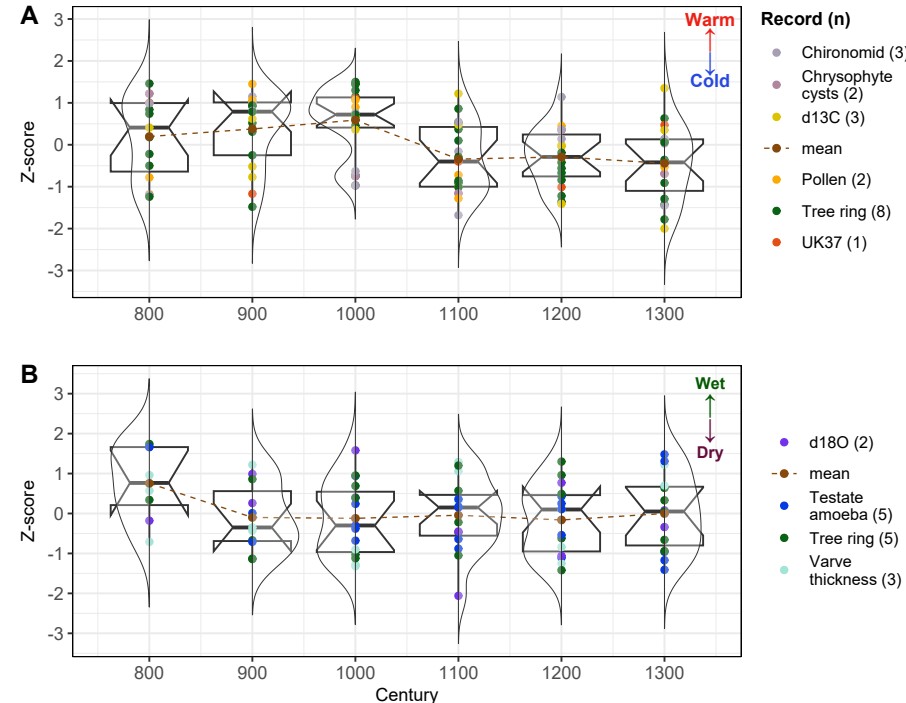

**Figure 4.** Overview of temperature (A) and P-E (B) conditions over Europe. The 'Record (n)' denotes the proxy and the number of proxies employed for the analysis.

In North America, continental-scale temperatures present a different climate trend compared to Europe: from the $9^{th}$ to the late $11^{th}$ centuries, average conditions were warm, followed by the onset of cold climate from the $12^{th}$ century (Figure

5-A). Concerning precipitation, from the $9^{th}$ to the late $11^{th}$ century, average hydrological conditions appeared dry, with the precipitation remaining slightly below their centennial mean (Figure 5-B). However, since the $12^{th}$ century, hydrological conditions appear to be normal, i.e., equal to the centennial mean, while from the $13^{th}$ to $14^{th}$ century, there are signals of wet conditions.

The analysis of temperature CIs (Figure 5-A) indicates consistent temperature conditions from the $8^{th}$ to the $13^{th}$ century.

However, a distinct shift in temperature is observed in the $14^{th}$ century, suggesting a transition towards colder conditions. Notably, the CIs for this century do not overlap with those of preceding centuries, underscoring a statistically significant change in temperature. For P-E records (Figure 5-B), the CIs around the mean indicate general stability with some variability from the $8^{th}$ to the $13^{th}$ century. However, a significant shift occurs in the $14^{th}$ century, indicating drier conditions, as evidenced by the non-overlapping CIs compared to previous centuries. This transition is marked by a pronounced shift in the P-E distribution





towards more negative values, suggesting a significant decrease in P-E availability. This shift indicates a prolonged drying trend that aligns with broader regional climatic changes. The significant change in P-E conditions is underscored by the non-overlapping CIs, reflecting a clear and statistically significant shift in the climatic regime. Overall, the CIs for temperature and P-E conditions in both Europe and North America reveal substantial climatic fluctuations during the $14^{th}$ century. These fluctuations are marked by pronounced cooling and drying trends.

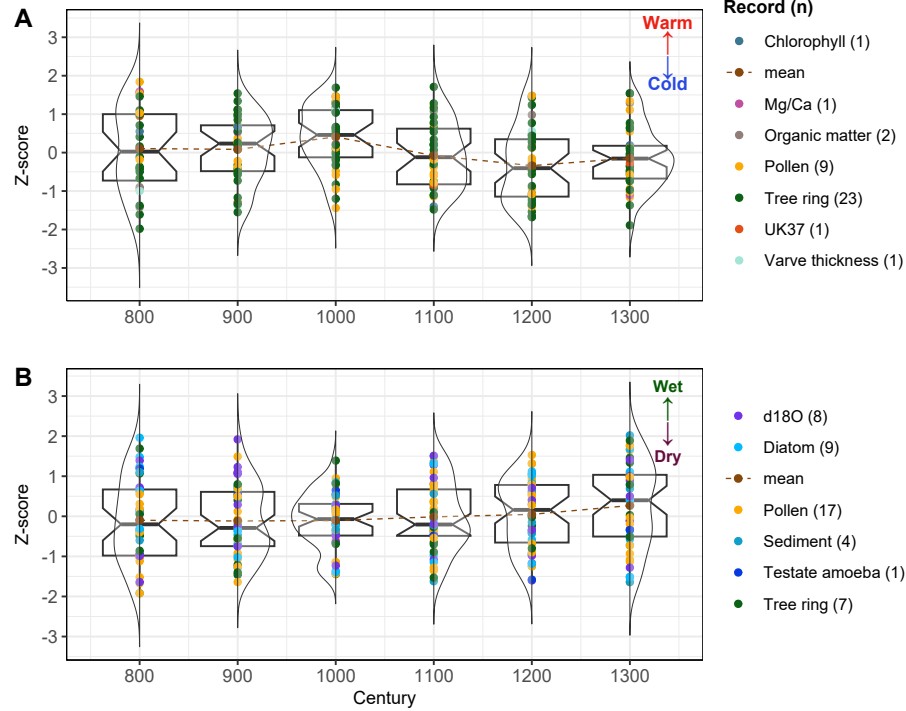

**Figure 5.** Overall view temperature (A) and P-E (B) conditions over North America. The 'Record (n)' denotes the proxy and the number of proxies employed for the analysis.

## 230 3.3 Spatiotemporal patterns of hydroclimates

Spatiotemporal analysis over Europe indicates that most regions experienced warm conditions during the $10^{th}$ and $11^{th}$ centuries. This warm climate was followed by colder climates from the $12^{th}$ to the $14^{th}$ centuries and wet conditions during the $9^{th}$ century. However, we observed heterogeneous signals of P-E across subsequent centuries at the centennial scale, potentially attributable to spatial heterogeneity (Figure 6). Notably, our results highlight the $11^{th}$ century as a period characterized

by relatively higher temperatures of MCA. A shift towards colder conditions commenced in the $12^{th}$ century, prevailing across Europe throughout the $12^{th}$ to $14^{th}$ centuries.

Regarding hydroclimate shifts across Europe, our findings unveil a notable latitudinal shift in temperature patterns, particularly from the $12^{th}$ to the $14^{th}$ centuries (Figure 6-A). This shift is characterized by the migration of cold climates from





high-latitudes to mid-latitudes. As a result of this latitudinal temperature shift, during the $14^{th}$ century, most central European

regions became even colder than in previous centuries. During the transition from the MCA to the LIA, specifically between the $13^{th}$ and $14^{th}$ centuries, the majority of northern and south-western regions, on average, experienced cold conditions. However, hydrological conditions reveals a complex hydroclimatic scenario during this transitional period. Notably, over Central Europe and a few southern areas like the northern Aegean region shifted toward humid conditions (Figure 6-B). During the central period of the MCA, specifically the $10^{th}$ and $11^{th}$ centuries, our analysis demonstrates a warm climate. Nevertheless,

in terms of regional hydrological conditions, there is spatial variability, with certain regions experiencing wet conditions while others depict arid conditions.

In summary, our observations suggest that during the MCA, coastal regions of Europe mostly experienced humid conditions, except for the $10^{th}$ and $11^{th}$ centuries, while inland areas were predominantly arid. For the transition from the MCA-LIA, our findings uncover a complex hydroclimatic scenario. While some records suggest a cold and arid climate in northern areas,

contradictory evidence points to a cold and humid climate regime in southern and western regions.







**Figure 6.** Proxy-estimated temperature (A) and P-E (B) conditions across Europe. The color scale values represent the z-scores.



For North America, the analysis of temperature distributions in high and mid-latitude regions revealed a predominantly warm climate during the $9^{th}$ and $11^{th}$ centuries (Figure 7), albeit with some regions displaying cold conditions as well. Commencing from the $12^{th}$ century, a transition towards cooler climate began prevailing in high and mid-latitude regions. Initially, this change was most pronounced in coastal areas and gradually extended further inland. More specifically, the onset of cold climates was observed in eastern North America and gradually extended to central and western regions between the $12^{th}$ to the $14^{th}$ centuries. In our examination of regional-scale hydroclimate variability, we noted a recurring cyclical pattern of dry and wet conditions at a centennial scale in the northern part of British Columbia. Dry conditions were evident in the $9^{th}$, $11^{th}$, and $13^{th}$ centuries, while wet conditions prevailed during the $10^{th}$, $12^{th}$, and $14^{th}$ centuries. In certain regions like the north of Alberta, our observations indicated a dry climate when the temperature was warm but close to its centennial mean. However, as the climate became even warmer, these areas exhibited a wetter climate.

Hydrological conditions in high and mid-latitude regions exhibited spatial and temporal heterogeneity, with some areas showing humid responses to warm conditions while others experienced arid conditions. This complex spatial pattern of hydrological response persisted during the cold climate phases, with most regions being arid and some remaining humid. Overall, hydroclimate patterns during the MCA in North America included all the possible hydroclimatic combinations, i.e., as warm-humid, warm-arid, cold-humid, and cold-arid regimes. Our findings imply that arid hydrological conditions in North American regions coincided with periods when temperatures exceeded their centennial-scale averages, whereas humid conditions were more prevalent when temperatures were close to or aligned with their centennial-scale averages.





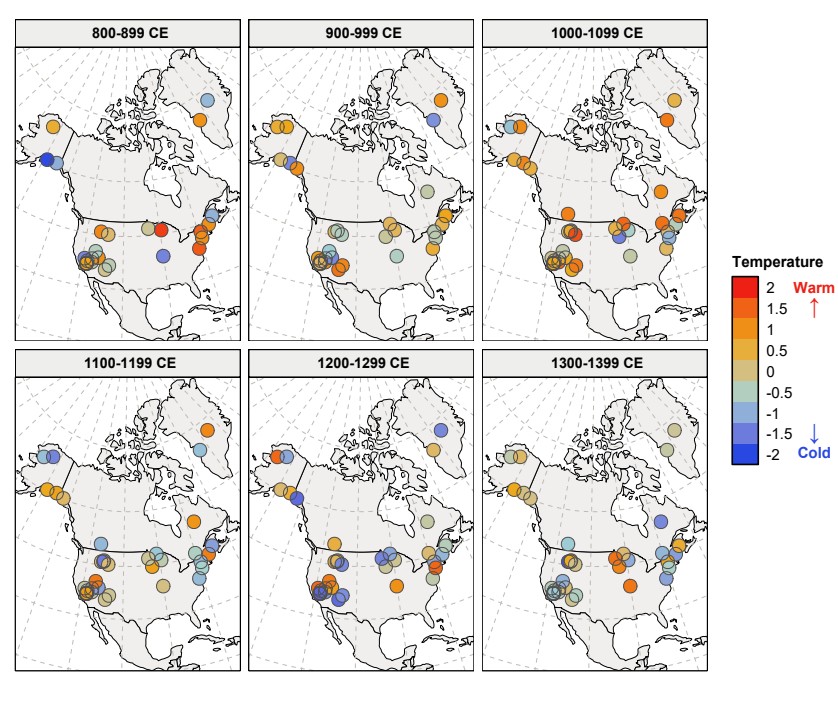

A.

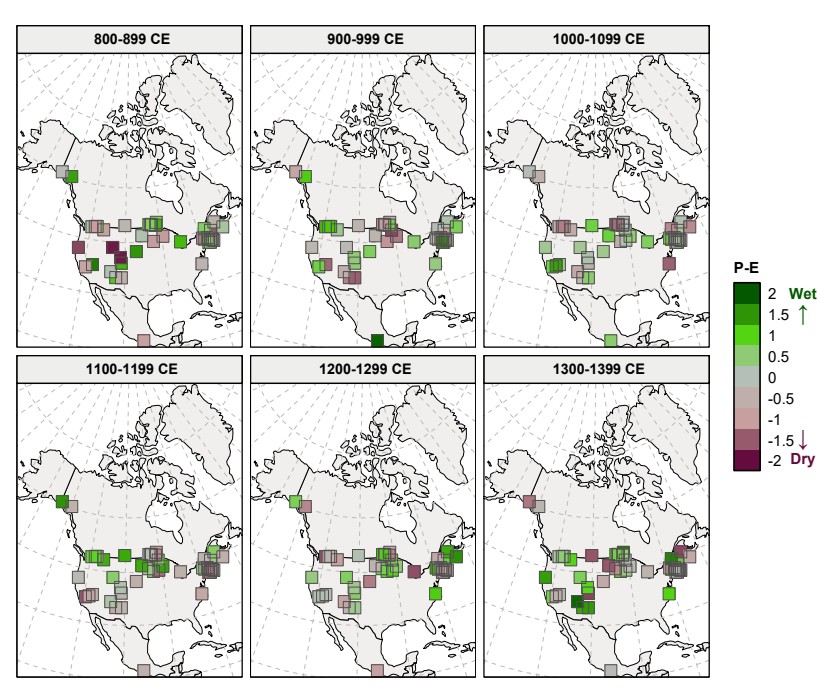

B.

**Figure 7.** Overview of temperature (A) and P-E (B) conditions over North America. The color scale values represent the z-scores.





### 3.3.1 Comparative assessment of outcomes from proxy and model assimilation

The comparison between proxy and model outputs offers valuable insights for accurately constraining future hydroclimatic
conditions using climate model projections. Therefore, this study compared proxy results with model assimilations, specifically
PHYDA and Paleoview, as shown in Figure 8(A, B). We categorised the model assimilations into two types of outputs: point
and grid. The point output represents values at the precise locations of the proxy data, while the grid output presents average
conditions across the continent.

We observed that both PHYDA and Paleoview assimilations indicate warmer climatic conditions from $9^{th}$ to mid-$11^{th}$
century, while PHYDA is overestimating the temperature conditions over a continental scale at grid scale (Figure 8). Our
analysis shows that temperatures provided by Paleoview both at grid and point levels across high and mid-latitudes align
closely with the trends observed in proxy records. In contrast, temperatures modelled by PHYDA align well with proxy-data at
grid scale, but opposing trend beginning from the $9^{th}$ century in high-latitudes are observed at the point scales. At mid-latitudes,
both PHYDA grid and point data show an underestimation of temperature compared to the proxy records.

Both the proxy data and model assimilation (particularly the PHYDA grid and Paleoview point) indicated predominantly arid
conditions at high-latitudes. This aridity is particularly notable from the mid-$10^{th}$ to the late-$11^{th}$ century in the PHYDA grid
assimilation and extends until the mid-$13^{th}$ century in the Paleoview point assimilation. However, discrepancies arise, with both
PHYDA point- and Paleoview grid-scales showing humid conditions at high- and mid-latitudes from mid-$10^{th}$ to the mid-$11^{th}$.
Additionally, at mid-latitudes, differences between the proxy and PHYDA assimilation become evident. PHYDA assimilation
suggest humid conditions from the $10^{th}$ to mid-$12^{th}$ century, while the proxy data suggests arid conditions. Simultaneously,
Paleoview is showing an underestimation for arid conditions.

In summary, our comparison reveals divergent trends in model assimilation under similar climate conditions. Paleoview
indicates that warm climates tend to promote humid conditions at high and mid-latitudes. PHYDA, on the other hand, suggests
that warm climates result in arid conditions at high-latitudes and humid conditions at mid-latitudes. At the grid scale, both
models generally satisfactorily reproduce climatic conditions at mid and high-latitudes. However, at regional scale, some
discrepancies arise, possibly due to model biases or high uncertainty and/or limited availability of data for model assimilation
in those specific regions.

**Climate**
**of the Past**
Discussions

**Europe**



**Figure 8.** Evaluation of model (Paleoview (A), PHYDA (B)) and proxy estimated precipitation (P-E) and temperature (Temp) conditions over Europe. The model point represents the exact location of proxy data.

Figure 9 presents the results of our temporal variability assessment using both proxy data and model assimilation over North America. At point scale, both models exhibit a pattern similar to proxy records. However, the models either over- or underestimate temperature and precipitation. Data-Model discrepancies become apparent, particularly at grid scales. For instance, lower temperatures and higher precipitation are observed in Northeastern America from the $9^{th}$ to the $12^{th}$ century. By contrast, western regions exhibit lower temperatures and lower precipitation during this period. After the $12^{th}$ century, lower/higher temperatures are associated with reduced precipitation over both eastern and western North America. At point





scale, model output show a modest agreement with proxy data. Temperature simulated by both PHYDA and Paleoview in
eastern and western North America align with the proxy data, with higher temperatures during the $10^{th}$ to late $11^{th}$ century.

However, we noticed that both proxy data and model assimilation depict a shift from a warm to a cold state, specifically
after the $11^{th}$ century. In terms of precipitation, both PHYDA and Paleoview diverge from the proxy data. While the proxy
data suggests below-average (centennial-scale) precipitation between the $10^{th}$ to late $11^{th}$ centuries, both model assimilation
indicate an increase during this period. In some cases, the proxy and model point data align well, whereas the model results
at the grid scale deviate from data estimates. This discrepancy underscores the presence of substantial uncertainties in the
proxy and model point data, potentially arising from limited data availability or measurement errors. The notable uncertainty
in the model assimilation highlights the importance of refining the models and reducing measurement errors to enhance our
understanding and prediction of hydroclimate variability.




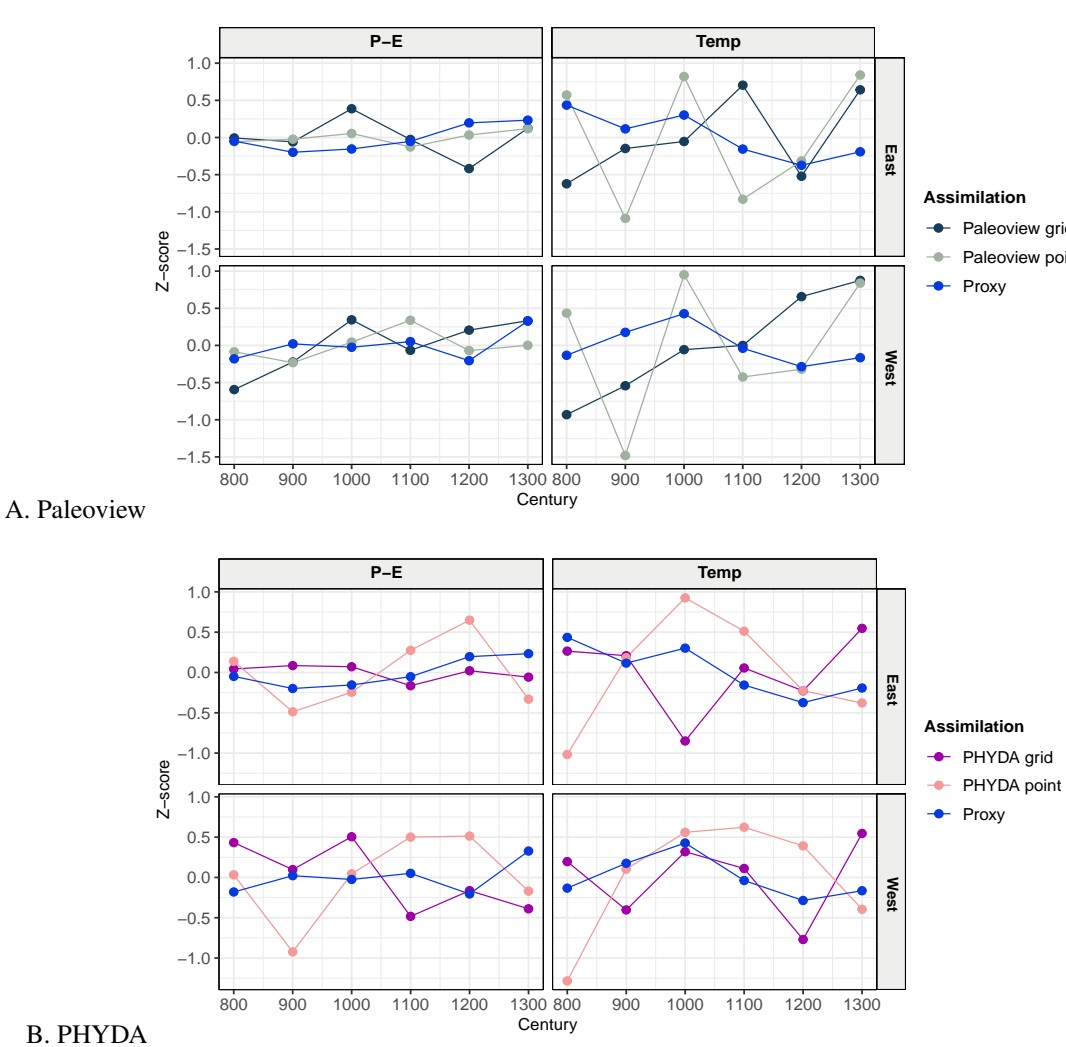

**Figure 9.** Evaluation of model (Paleoview (A), PHYDA (B)) and proxy estimated precipitation (P-E) and temperature (Temp) conditions over North America. The model point represents the exact location of proxy data.

The comparison between model assimilations and proxy data reveals variability in correlations depending on the variable
and region (Table 1). For P-E, a significant/strong correlation is observed between PHYDA points and proxy data over Europe, while all other comparisons exhibit low/non-significant correlations. For temperature, correlations are generally low/non-significant across all comparisons. For North America, both P-E and temperature variables show non-significant correlations across all proxy and model comparisons. These results underscore the challenge of achieving better agreement between model assimilations and proxy data, highlighting the need for improvements in model performance.





**Table 1.** Correlations between proxy and model assimilations in terms of $R^2$ and p-values.

| Continent | Comparison | Variable | $R^2$ | p value |
|---|---|---|---|---|
| Europe | Proxy vs PHYDA grid | P-E | 0.09 | 0.35 |
| Europe | Proxy vs PHYDA point | P-E | 0.68 | ∼0 |
| Europe | Proxy vs Paleoview grid | P-E | 0.06 | 0.44 |
| Europe | Proxy vs Paleoview point | P-E | ∼0 | 0.94 |
| Europe | Proxy vs PHYDA grid | Temp | 0.19 | 0.15 |
| Europe | Proxy vs PHYDA point | Temp | ∼0 | 0.99 |
| Europe | Proxy vs Paleoview grid | Temp | 0.05 | 0.49 |
| Europe | Proxy vs Paleoview point | Temp | 0.09 | 0.34 |
| N.America | Proxy vs PHYDA grid | P-E | 0.18 | ∼0 |
| N.America | Proxy vs PHYDA point | P-E | ∼0 | 0.99 |
| N.America | Proxy vs Paleoview grid | P-E | 0.01 | 0.77 |
| N.America | Proxy vs Paleoview point | P-E | 0.09 | 0.36 |
| N.America | Proxy vs PHYDA grid | Temp | ∼0 | 0.97 |
| N.America | Proxy vs PHYDA point | Temp | 0.02 | 0.66 |
| N.America | Proxy vs Paleoview grid | Temp | 0.14 | 0.24 |
| N.America | Proxy vs Paleoview point | Temp | 0.04 | 0.54 |

## 4 Discussion

### 4.1 State of the North Atlantic during the MCA

The state of the North Atlantic plays a pivotal role in shaping hydroclimate patterns across globe. In this study, we focused on key factors including the AMOC and SST conditions during the MCA period to characterize the state of the North Atlantic. Various modelling studies have suggested a weakened response of the AMOC to warm climate conditions (Dima et al., 2021; Cini et al., 2024). However, our investigation did not find any evidence of AMOC weakening in response during the warmer MCA (Latif et al., 2022). Although we selected various (indirect) AMOC proxies from different locations across the North Atlantic Ocean, determining the AMOC condition during the warm climate remains challenging. It is possible that the AMOC underwent changes during that time, but the signal may have been too weak to be captured by the selected proxies. The proxy-records we selected exhibit two distinct responses. Six records show a seemingly weaker AMOC during the $10^{th}$ and $11^{th}$ century, while two records show the exact opposite. Even within similar proxies (e.g., sortable silts, Figure 2.C), the data-spread is very large and prevents us to draw any clear picture of temporal AMOC changes during the MCA. Definitive conclusions on the sensitivity of AMOC tracers across both space and time require further investigation.

Sea Surface Temperatures (SSTs) (Figure 3.A) were consistently higher over the North Atlantic region from the $9^{th}$ to the mid of $10^{th}$ century. Low SST are observed during the mid-$10^{th}$ to $14^{th}$ centuries, which may be related to a strengthening of the Subtropical high-pressure system over the North Atlantic (Gibson et al., 2024). In addition, freshwater discharge into the North Atlantic could have also played a role in reducing the SST in the region (Schmidt et al., 2004; Levy et al., 2023). Low SST in the North Atlantic Ocean may have triggered a shift of the ITCZ towards the southern hemisphere (Steinman et al., 2022;





Giannini et al., 2000), particularly during the $10^{th}$ and $11^{th}$ centuries. These oceanic influences likely reshaped atmospheric circulation and dynamics over the North Atlantic region, expanding colder climate regions and constraining moisture flux

across North America and Europe (Frankignoul and Kestenare, 2005), in particular between the $12^{th}$ and $14^{th}$ centuries.

## 4.2 Relationships between SSTs and hydroclimates in the north Atlantic region

The variations in SSTs in the North Atlantic Ocean are known to significantly influence regional climates in Europe and North America (Czaja and Frankignoul, 2002; Jones and Anderson, 2008). We observed a link between the warm climate of Europe and high North Atlantic SST, particularly during the onset of the MCA. It is noticed that high SSTs are potentially linked with

a positive phase of the North Atlantic Oscillation (NAO) and intensified westerlies (Delworth and Greatbatch, 2000; Hurrell and Deser, 2010; Kim et al., 2023). It is plausible that the positive phase of the NAO and enhanced westerlies facilitated the redistribution of atmospheric heat across both continents, thereby contributing to the warm conditions observed (Trouet et al., 2009). However, we observed that as SST began to decrease from the mid-$10^{th}$ century, transitioning to a lower state, the warm climate also shifted towards a colder climate across Europe and North America. This decline in SST, hindering northward

meridional heat flow, further contributed to a weakened NAO and westerlies (Lehner et al., 2013).

Furthermore, this variability in SST and westerlies would have played a significant role in European hydroclimate variability. The low (high) SST influenced the transport of cold (warm) air by weaker (stronger) westerlies to western and southern European regions, thereby causing cold (warm) and arid (humid) climates (Trouet et al., 2009), respectively. We speculate that the cooling of the North Atlantic Ocean observed from the mid-$10^{th}$ to the $14^{th}$ century, likely due to low SST, led to

a reduction in moisture flux and decreased temperatures towards European regions. This decrease in moisture flux from the ocean source and consequent cold climate may have contributed a spatially diverse pattern of precipitation across Europe, characterized by both wet and arid conditions (Liu et al., 2020). However, it is important to note that this spatially diverse pattern of precipitation across Europe may have also been influenced by regional thermodynamic responses due to changes in ocean-atmospheric circulation (Pratap and Markonis, 2022).

Besides Europe, hydroclimate variability in the North American region likely responded to centennial-scale fluctuations in SST (Cook et al., 2022). The warm conditions observed over North America during the MCA may be associated to intensified heat transport facilitated by strengthened westerlies, influenced by the high North Atlantic SST condition. Furthermore, during the $9^{th}$ to mid-$10^{th}$ centuries, the high SST may have contributed to a positive feedback mechanism (water vapour feedback). These high SSTs and the resulting positive feedback mechanism conditions likely led to arid or drought-like conditions in

western and southern North America (Seager et al., 2007), while the northwestern regions experienced increased humidity during that period. The arid conditions are likely linked to episodes of drought, which have also been observed and documented by Cook et al. (2014) and Chen et al. (2021).

However, as discussed earlier, by the mid-$10^{th}$ century, the SST began decreasing, leading a reduced atmospheric moisture supply to eastern and other parts of North America. This, in turn, led to reduced precipitation and drought-like conditions in

these regions (Seager et al., 2008). Mainly, during the MCA, western and central regions of North America experienced warm and arid climatic conditions, characterized by decadal or centennial-scale megadrought events (Oglesby et al., 2011). While





the underlying factors behind these megadrought events are still under investigation, our findings suggest that these drought conditions in North America may have been linked to a southward shifting ITCZ in response to low SST. This shift likely constrained the moisture supply over these regions.

The precise forcing of hydroclimates during the MCA in North America remain subject to ongoing debate. Notably, the climatic conditions in North America appear significantly influenced by North Atlantic Ocean fluctuations (Hurrell and Deser, 2010). Our study also suggests an agreement on a relationship between warm North Atlantic Ocean conditions and hydroclimate variability over North America. Proxy observations revealed spatially distinct precipitation patterns across North America. We found that coastal areas generally experienced increased precipitation when temperatures were close to their centennial-

scale means, while arid climates were associated with temperatures exceeding this mean. By contrast, more inland regions experienced higher precipitation when temperatures were high. We show that both in eastern and western North America, lower (higher) temperatures were associated with more arid (humid) conditions. These results suggest that variations in temperature influenced precipitation patterns across the continent.

### 4.3   Potential drivers of hydroclimate changes over North America and Europe

Temperature and precipitation are integral components of the hydroclimate system. Many studies have emphasized the high sensitivity of hydroclimate patterns in Europe to fluctuations in North Atlantic oceanic circulation, with documented occurrences of recurrent drought-like conditions during the MCA (Helama et al., 2009; Hernández et al., 2020). However, the validation and reliability of hydroclimatic records from the MCA in Europe, particularly regarding the origins of drought-like conditions, are uncertain and sometimes contradictory, necessitating further verification (Seager et al., 2007; Carrillo et al.,

2022). We hypothesize that the warm climate of the MCA might have resulted in increased temperatures in the Arctic regions (Mann et al., 2009), potentially enabling the influx of atmospheric moisture. This, in turn, could have resulted in a humid climate in northwest Europe, particularly during the $11^{th}$ and $13^{th}$ centuries. On the other hand, the prolonged warm climate spanning over three centuries (from the $9^{th}$ to the $11^{th}$ centuries) likely exerted a significant influence on the climate dynamics of the Arctic regions (Bindoff et al., 2013). This led to the melting of ice sheets, glaciers, and sea ice, resulting in a substantial

influx of freshwater into the North Atlantic Ocean. This change over the North Atlantic Ocean potentially established a linkage between the MCA-LIA transition and hydroclimate fluctuations across North America and Europe (Holliday et al., 2020).

The influx of low-density freshwater may have impacted deep ocean convection by reducing density in the Labrador Sea, resulting in the cooling and freshening of the North Atlantic Current (Robson et al., 2014). These changes in the North Atlantic Ocean notably regulated the state of the SST and the position of the ITCZ in the North Atlantic region. Specifically, cooling

and freshening led to low SST, which likely characterized the emergence of the MCA-LIA transition (Trouet et al., 2012). Most likely, the low SST over the North Atlantic ocean would have constrained the northward movement of warm tropical water, resulting in the development of an SST gradient (Bond et al., 2001). Consequently, the increased SST gradient over the North Atlantic probably restricted the transport of atmospheric heat and moisture from the tropics (Srokosz and Bryden, 2015), contributing to the southward shift of the ITCZ (Giannini et al., 2000). On the other hand, the abrupt cold climate that





occurred after the $13^{th}$ century, known as the LIA, may have been a response to the prolonged low SST and southward ITCZ (Lynch-Stieglitz, 2017).

Notably, during the warm phases of the MCA, we observed warm conditions coinciding with high SST and a northward shift in the ITCZ. Throughout this period, continental-scale hydroclimate displayed warm and wet conditions across both continents. However, after the peak warmth of the MCA (i.e., after $11^{th}$ century), the state of SST and ITCZ underwent divergent

shifts–transitioning from high to low and northward to southward, respectively. Concurrently, continental-scale hydroclimate patterns over North America and Europe also shifted. Notably, after $11^{th}$ century, temperatures shifted from warm to cold, and hydrological conditions transitioned from wet to dry. In the meantime, at a regional scale across both continents, temperature changes appeared associated with shifts in the North Atlantic state, while precipitation patterns remain uncertain. Regional precipitation distribution seems more linked to specific changes in the local climate system, particularly in response

to thermodynamic shifts. In specific, the occurrence of humid or arid conditions in certain locations across Europe and North America could be associated with localized responses to amplified regional atmospheric warming (Tabari, 2020). However, further investigations are needed to confirm the precise response of hydrological conditions in a warm climate.

In summary, our observations indicate that variations in SST and ITCZ exert a significant influence on hydroclimate distributions, as the variability in both appears to shift in parallel. These variations in the North Atlantic state impact ocean-

atmospheric-land circulation patterns, thereby constraining the distribution of terrestrial moisture and heat, which contribute to shaping the hydroclimate.

## 4.4    Proxy and model assimilation comparison

When comparing model assimilation (PHYDA and Paleoview) with proxy outcomes, we observed significant discrepancies between them. The uncertainties in both proxy and model outcomes may depend on several key factors. For instance, proxies

are often spatially unevenly distributed and capture localized climate variations, which may not fully represent broader climate dynamics. Conversely, models introduce biases through the simplification of physical processes and assumptions embedded in parameterizations, especially when simulating complex regional dynamics. These limitations complicate direct comparisons of proxy with climate models, which simulate conditions over comprehensive grids.

On the other hand, the discrepancies between grid- and point-based outcomes over both European and North American

regions may also be attributed to several factors. For instance, climate model grids average climate conditions over large areas, which can mask the fine-scale processes that are captured well by regional proxies. In Europe, specially over high- and mid-latitude regions temperature and precipitation distribution are significantly influenced by different climatic drivers, such as polar jet streams and mid-latitude cyclones, which may be inadequately represented in models but more accurately reflected in proxy records (Woollings, 2010; Studholme et al., 2022). Similarly, temperature and precipitation variability over the east-

west contrast in North America arises from distinct climatic regimes: the west is dominated more by oceanic influences and orographic effects, while the east experiences greater continental variability (Trenberth et al., 2007). Proxies capture these localized signals, whereas models often generalize them, which can lead to discrepancies in regional climate dynamics.



In conclusion, our findings reveal that while both models generally align with proxy trends at the grid scale, notable discrepancies emerge at the regional level. Paleoview tends to associate warmer climates with humid conditions, while PHYDA
indicates arid conditions at high latitudes. These divergent trends underscore the importance of careful model selection and the need for improved data availability to enhance future hydroclimatic projections.

## 5    Conclusions

This study investigates centennial-scale hydroclimate patterns across Europe and North America during the MCA, aiming to uncover potential connections between these patterns and the North Atlantic variability (in SST, AMOC, and the ITCZ) within
a warm climate context. Although the state of the AMOC during warm climates remains uncertain, we found that centennial-scale variability in SST and ITCZ dynamics played a pivotal role in shaping hydroclimate patterns across both continents and in the transitional shift from the MCA to the LIA. This transitional period witnessed a shift in hydroclimate across North America and Europe, transitioning from warm to cold and wet to dry climates, with persistent uncertainties in hydrological conditions. During the warm climate, hydroclimate changes and their link with the North Atlantic variability demonstrate a
lead-lag relationship, notably prominent in temperature variations. Our interpretation of a lead-lag relationship is rooted in the alignment between a warm climate and a strengthened North Atlantic condition, which later transitioned to weaker states, potentially due to increased input of freshwater from Arctic melting and sea ice.

We anticipate that variations in the North Atlantic, including SST and the ITCZ, may have acted as possible factors in altering hydroclimate patterns at the continental scale. Specifically, low SST and southward shifting ITCZ. This led to cold and
arid hydroclimate conditions across Europe and North America, particularly after the $11^{th}$ century. This transition may have played a significant role in initiating the LIA (Trouet et al., 2009; Wanamaker et al., 2012). When considering the interplay between temperature and precipitation, it is crucial to acknowledge the complexity and spatial variability of their relationship. Our study provides a comprehensive overview, revealing that during the MCA, elevated temperatures exceeding the centennial scale average were generally associated with arid climates. Conversely, slightly warmer conditions than the centennial average
tended to coincide with more frequent occurrences of humid conditions. Additionally, our investigation observed that increased precipitation under low temperatures was sporadic, possibly indicating regional climate heterogeneity.

Overall, our findings underscore the significant modulation of hydroclimate variability during the MCA due to North Atlantic variability, particularly in SST and shifts in the ITCZ. This North Atlantic modulation influenced the redistribution of atmospheric heat and water vapor, thereby shaping the hydroclimate across the Northern Hemisphere. Based on the evidence
from the MCA, it is plausible to consider that ongoing climate warming may contribute to the future lowering in SST and a southward shift in ITCZ. Consequently, this may result in a similar hydroclimate shift as observed during the MCA. However, the intricate relationship between hydrological variability and its latitudinal heterogeneity, particularly under warmer climate conditions, remains enigmatic and warrants further in-depth investigation.



*Data availability.* The data used in this study are open-source and publicly available on the website of NOAA's National Centers for Environ-
mental Information (https://www.ncei.noaa.gov/products/paleoclimatology) and the Past Global Changes (PAGES) (https://pastglobalchanges.
org/science/data/databases).

## Appendix A: Data tables

**Table A1.** Source and distribution information for precipitation (Precip) and temperature (Temp) across the European region.

| Region | Material | Fractions | Proxy | Variable | Age uncertainty (years) | Latitude | Longitude | Investigators | Publication |
|---|---|---|---|---|---|---|---|---|---|
| North Aegean | Tree | Tree-rings | Total ring width | Precip | | 40.5 | 29.5 | Griggs et al., 2007 | https://doi.org/10.1002/joc.1459 |
| Central and Eastern Pyrenees | Lake sediment | Sediment | Chrysophyte cyst | Temp | | 41.5 | 0.75 | Pla et al., 2004 | https://doi.org/10.1007/s00382-004-0482-1 |
| Lake Redon | Lake sediment | Sediment | Chrysophyte cyst | Temp | | 42.64 | 0.77 | Pla et al., 2004 | https://doi.org/10.1007/s00382-004-0482-1 |
| Northern inland Iberia | Stalagmite | Stalagmite | $\delta^{13}$C | Temp | ± 5 | 42.69 | -3.94 | Martín-Chivelet et al., 2011 | https://doi.org/10.1016/j.gloplacha.2011.02.002 |
| Lake Allos | Lake sediment | Sediment | Varve thickness | Precip | ± 30 | 44.25 | 6.72 | Wilhelm et al., 2012 | https://doi.org/10.1016/j.yqres.2012.03.003 |
| Seebergsee | Lake sediment | Chironomid | C-assemblage | Temp | ±12 | 46.12 | 7.47 | Larocque et al., 2012 | https://doi.org/10.1016/j.quascirev.2012.03.010 |
| Lake Oeschinen | Lake sediment | Sediment | Varve thickness | Precip | ± 9 | 46.5 | 7.73 | Amann et al., 2015 | https://doi.org/10.1016/j.quascirev.2015.03.002 |
| Alps | Tree | Tree-rings | Total ring width | Temp | | 47.5 | 12.5 | Büntgen et al., 2016 | https://doi.org/10.1038/ngeo2652 |
| Upper Rhine Valley | Tree | Tree-rings | Total ring width | Precip | | 48.75 | 7.75 | Tegel et al., 2020 | https://doi.org/10.1038/s41598-020-73383-8 |
| Tatra region | Tree | Tree-rings | Total ring width | Temp | | 49.5 | 20 | Büntgen et al., 2013 | https://doi.org/10.1073/pnas.1211485110 |
| Main River Region | Tree | Tree-rings | Total ring width | Precip | | 50.05 | 10.2 | Land et al., 2019 | https://doi.org/10.5194/cp-15-1677-2019 |
| Meerfelder Maar | Lake sediment | Pollen | P-assemblage | Temp | | 50.1 | 5.5 | Litt et al., 2009 | https://doi.org/10.1111/j.1502-3885.2009.00096.x |
| Holzmaar | Lake sediment | Pollen | P-assemblage | Temp | ± 5 | 50.12 | 6.88 | Litt et al., 2009 | https://doi.org/10.1111/j.1502-3885.2009.00096.x |
| Southern-Central England | Tree | Tree-rings | Total ring width | Precip | | 51.6 | -1.75 | Wilson et al., 2012 | https://doi.org/10.1007/s00382-012-1318-z |
| East Anglia | Tree | Tree-rings | Total ring width | Precip | | 52.5 | 1 | Cooper et al., 2012 | https://doi.org/10.1007/s00382-012-1328-x |
| Stazki Bog | Peatland sediment | Sediment | Testate amoeba | Precip | ± 30 | 53.42 | 18.08 | Lamentowicz et al., 2010 | https://doi.org/10.1016/j.palaeo.2008.04.023 |
| Ireland | Peatland sediment | Sediment | Testate amoeba | Precip | | 53.5 | -8 | Swindles et al., 2013 | https://doi.org/10.1016/j.earscirev.2013.08.012 |
| Slowinskie Blota | Peatland sediment | Sediment | Testate amoeba | Precip | ± 30 | 54.36 | 16.49 | Lamentowicz et al., 2010 | https://doi.org/10.1111/j.1502-3885.2008.00047.x |
| Stazki Bog | Peatland sediment | Sediment | Testate amoeba | Precip | ± 30 | 54.42 | 20.08 | Lamentowicz et al., 2010 | https://doi.org/10.1016/j.palaeo.2008.04.023 |
| Northern Britain | Peatland sediment | Sediment | Testate amoeba | Precip | | 55.5 | -2.75 | Charman et al., 2006 | https://doi.org/10.1016/j.quascirev.2005.05.005 |
| Uamh an Tartair | Stalagmite | Stalagmite | $\delta^{18}$O | Precip | | 58.15 | -5.98 | Baker et al., 2010 | https://doi.org/10.1016/j.gloplacha.2010.12.007 |
| Finnish Lakeland | Tree | Tree-rings | Total ring width | Temp | | 62.33 | 28.33 | Helama et al., 2014 | https://doi.org/10.2478/s13386-013-0163-0 |
| Jämtland | Tree | Tree-rings | Total ring width | Temp | | 63 | 14.05 | Zhang et al., 2016 | https://doi.org/10.5194/cp-12-1297-2016 |
| Stora Vidarvatn | Lake sediment | Chironomid | C-assemblage | Temp | | 66.24 | -15.83 | Axford et al., 2008 | https://doi.org/10.1007/s10933-008-9251-1 |
| Northern Fennoscandia | Tree | Tree-rings | Total ring width | Temp | ± 15 | 68.15 | 24.38 | Matskovsky and Helama, 2014 | https://doi.org/10.5194/cp-10-1473-2014 |
| Torneträsk | Tree | Cellulose | $\delta^{13}$C | Temp | | 68.2 | 19.8 | Loader et al., 2013 | https://doi.org/10.1016/j.quascirev.2012.11.014 |
| Torneträsk | Tree | Tree-rings | Total ring width | Temp | ± 15 | 68.26 | 19.63 | Melvin et al., 2013 | https://doi.org/10.1177/0959683612460791 |
| Laanila | Tree | Tree-rings | Total ring width | Temp | | 68.5 | 27.5 | Gagen et al., 2011 | https://doi.org/10.1029/2010GL046216 |
| Northern Fennoscandia | Tree | Tree-rings | Total ring width | Temp | | 68.63 | 24.69 | Helama et al., 2012 | https://doi.org/10.1007/s10933-012-9598-1 |
| Forfjorddalen | Tree | Cellulose | $\delta^{13}$C | Temp | | 68.8 | 15.73 | Young et al., 2012 | https://doi.org/10.1007/s00382-011-1246-3 |
| Kongressvatnet | Lake sediment | Sediment | $U^{k'}37$ | Temp | | 78.02 | 13.93 | Andrea et al., 2012 | https://doi.org/10.1130/G33365.1 |
| Lake Nuudsaku | Lake sediment | Sediment | $\delta^{18}$O | Precip | | 58.1969 | 25.6275 | Nathan et al., 2017 | https://doi.org/10.1016/j.quascirev.2017.09.013 |
| Lake Storsjön | Lake sediment | Varve | Varve thickness | Precip | | 63.12 | 14.37 | Labuhn et al., 2017 | https://doi.org/10.3390/quat1010002 |
| Kylmanlampi | Lake sediment | Chironomid | C-assemblage | Temp | | 64.3 | 30.25 | Luoto et al., 2003 | https://doi.org/10.3354/cr01331 |





**Table A2.** Source and distribution information for precipitation (Precip) and temperature (Temp) across the North American region. (Part-1)

| Region | Material | Fractions | Proxy | Variable | Age uncertainty (years) | Latitude | Longitude | Investigators | Publication |
|---|---|---|---|---|---|---|---|---|---|
| Albemarle Sound Drainage Basin | Tree | Tree-rings | Total ring width | Precip | | 36 | -77 | Stahle et al., 2011 | https://doi.org/10.1007/s12237-013-9643-y |
| Beaver Lake | Lake sediment | Diatom | D-assemblage | Precip | ±35 | 42.46 | -100.67 | Schmieder et al., 2011 | https://doi.org/10.1016/j.quascirev.2011.04.011 |
| Berry Pond | Lake sediment | Pollen | P-assemblage | Precip | | 42.51 | -73.32 | Marsicek et al., 2013 | https://doi.org/10.1016/j.quascirev.2013.09.001 |
| Blood Pond | Lake sediment | Pollen | P-assemblage | Precip | | 42.08 | -71.96 | Marsicek et al., 2013 | https://doi.org/10.1016/j.quascirev.2013.09.001 |
| Bufflehead Pond | Lake sediment | Sediment | GPR | Precip | ±40 | 44.99 | -93.54 | Shuman et al., 2009 | https://doi.org/10.1890/08-0985.1 |
| Castor Lake | Lake sediment | Sediment | $\delta^{18}O$ | Precip | | 48.54 | -119.56 | Steinman et al., 2012 | https://doi.org/10.1073/pnas.1201083109 |
| Chesapeake Bay | Marine sediment | Sediment | $Mg/Ca$ | Temp | | 38.61 | -76.4 | Cronin et al., 2010 | https://doi.org/10.1016/j.palaeo.2010.08.009 |
| Clear pond | Lake sediment | Pollen | P-assemblage | Temp, Precip | | 43 | -74 | Gajewski 1988 | https://doi.org/10.1016/0033-5894(88)90034-8 |
| Conroy lake | Lake sediment | Pollen | P-assemblage | Temp, Precip | | 46.17 | -67.53 | Gajewski 1988 | https://doi.org/10.1016/0033-5894(88)90034-8 |
| Dark Lake | Lake sediment | Pollen | P-assemblage | Precip | | 45.16 | -91.28 | Gajewski 1988 | https://doi.org/10.1016/0033-5894(88)90034-8 |
| Deep Lake | Lake sediment | Pollen | P-assemblage | Precip | | 41.56 | -70.64 | Marsicek et al., 2013 | https://doi.org/10.1016/j.quascirev.2013.09.001 |
| Dixie Lake | Lake sediment | Diatom | D-assemblage | Precip | | 49.83 | -93.95 | Laird et al., 2012 | https://doi.org/10.1111/j.1365-2486.2012.02740.x |
| El Malpais | Tree | Tree-rings | Total ring width | Precip | | 34.97 | -108.18 | Stahle et al., 2009 | https://doi.org/10.1175/2008JCLI2752.1 |
| El Malpais | Tree | Tree-rings | Total ring width | Precip | | 34.97 | -106.18 | Grissino and Henri 1995 | https://repository.arizona.edu/handle/10150/191192 |
| ELA Lake 239 | Lake sediment | Diatom | D-assemblage | Precip | | 49.67 | -93.73 | Laird et al., 2012 | https://doi.org/10.1111/j.1365-2486.2012.02740.x |
| ELA Lake 442 | Lake sediment | Diatom | D-assemblage | Precip | | 49.77 | -93.82 | Laird et al., 2012 | https://doi.org/10.1111/j.1365-2486.2012.02740.x |
| Emerald Lake | Lake sediment | Sediment | GPR | Precip | | 39.15 | -106.41 | Shuman et al., 2014 | https://doi.org/10.2113/gsrocky.49.1.33 |
| Étang_Fer-de-Lance | Lake sediment | Pollen | P-assemblage | Precip | ±80 | 45.36 | -72.23 | Claire et al., 2021 | https://doi.org/10.1177/0959683621994642 |
| Foy Lake | Lake sediment | Sediment | $\delta^{18}O$ | Precip | ±35 | 48.17 | -114.35 | Spruce et al., 2020 | https://doi.org/10.1016/j.qsa.2020.100013 |
| Fresh Pond | Lake sediment | Pollen | P-assemblage | Precip | | 41.16 | -71.58 | Jeremiah et al., 2013 | https://doi.org/10.1016/j.quascirev.2013.09.001 |
| Gall Lake | Lake sediment | Diatom | D-assemblage | Precip | | 50.23 | -91.45 | Kathleen et al., 2011 | https://doi.org/10.1111/j.1365-2486.2012.02740.x |
| Great Basin | Tree | Tree-rings | Total ring width | Temp | | 38 | -116.5 | Salzer et al., 2013 | https://doi.org/10.1007/s00382-013-1911-9 |
| Greenland-GISP2 | Ice core | Ice | d15N & d40Ar | Temp | | 72.6 | -38.5 | Kobashi et al., 2010 | https://doi.org/10.1007/s10584-009-9689-9 |
| Hell's Kitchen Lake | Lake sediment | Pollen | P-assemblage | Temp, Precip | | 46.11 | -89.42 | Gajewski 1988 | https://doi.org/10.1016/0033-5894(88)90034-8 |
| Horseshoe Lake (HORM12) | Lake sediment | Sediment | brGDGTs | Temp | | 38.7 | -90.08 | Munoz et al., 2020 | https://doi.org/10.1029/2020GL087237 |
| Iceberg Lake | Lake sediment | Sediment | Varve thickness | Temp | | 60.78 | -142.95 | Loso 2008 | https://doi.org/10.1007/s10933-008-9264-9 |
| Jellybean Lake | Lake sediment | Sediment | $\delta^{18}O$ | Precip | ±5 | 60.35 | -134.8 | Anderson et al., 2005 | https://doi.org/10.1016/j.yqres.2005.03.005 |
| Juxtlahuaca Cave | Stalagmite | Stalagmite | $\delta^{18}O$ | Precip | | 17.4 | -99.2 | Lachniet et al., 2017 | https://doi.org/10.1130/G32471.1 |
| Kurupa Lake | Lake sediment | Sediment | Chlorophyll | Temp | | 68.35 | -154.61 | Boldt et al., 2015 | https://doi.org/10.1177/0959683614565929 |
| L1_CANA458 | Tree | Tree-rings | Total ring width | Temp | | 54.21 | -71.35 | Bellen et al., 2019 | https://doi.org/10.1594/PANGAEA.905453 |
| Lake Braya Sø | Lake sediment | Sediment | $U^{k'}37$ | Temp | | 69.99 | -51.03 | Gunten et al., 2012 | https://doi.org/10.1038/srep00609 |
| Lake Mina | Lake sediment | Pollen | P-assemblage | Precip | | 45.89 | -95.48 | Jeannine et al., 2020 | https://doi.org/10.1016/j.quascirev.2008.01.005 |
| Lake of the Clouds | Lake sediment | Pollen | P-assemblage | Temp, Precip | | 48 | -91.07 | Gajewski 1988 | https://doi.org/10.1016/0033-5894(88)90034-8 |
| Lake of the Woods | Lake sediment | Sediment | GPR | Precip | | 43.48 | -109.89 | Pribyl and Shuman 2014 | https://doi.org/10.1016/j.yqres.2014.01.012 |
| Lime lake | Lake sediment | Sediment | $\delta^{18}O$ | Precip | | 48.87 | -117.34 | Steinman et al., 2012 | https://doi.org/10.1073/pnas.1201083109 |
| Little Pond | Lake sediment | Pollen | P-assemblage | Precip | | 42.68 | -72.19 | Marsicek et al., 2013 | https://doi.org/10.1016/j.quascirev.2013.09.001 |
| Little Raleigh | Lake sediment | Diatom | D-assemblage | Precip | | 49.45 | -91.89 | Kathleen et al., 2012 | https://doi.org/10.1111/j.1365-2486.2012.02740.x |
| Meekin Lake | Lake sediment | Diatom | D-assemblage | Precip | | 49.82 | -94.77 | Laird et al., 2012 | https://doi.org/10.1111/j.1365-2486.2012.02740.x |
| Minden Bog | Peatland sediment | Sediment | Testate amoeba | Precip | | 43.61 | -82.84 | Booth and Jackson, 2003 | https://doi.org/10.1191/0959683603hl669rp |
| Moon Lake | Lake sediment | Diatom | D-assemblage | Precip | | 46.86 | -98.16 | Laird et al., 1998 | https://doi.org/10.1038/384552a0 |
| Mt Logan | Ice core | Ice | $\delta^{18}O$ | Precip | | 60.58 | -140.5 | Fisher et al., 2008 | https://doi.org/10.1177/0959683608092236 |
| Nevada | Tree | Tree-rings | Total ring width | Precip | | 38 | -117 | Huges and Graumlich, 2000 | https://doi.org/10.1007/978-3-642-61113-1_6 |
| New Long Pond | Lake sediment | Sediment | GPR | Precip | ±40 | 41.85 | -70.68 | Newby et al., 2009 | https://doi.org/10.1016/j.quascirev.2009.02.020 |
| No Bottom Lake | Lake sediment | Sediment | GPR | Precip | | 41.29 | -70.11 | Marsicek et al., 2013 | https://doi.org/10.1016/j.quascirev.2013.09.001 |
| Oregon Caves | Stalagmite | Stalagmite | $\delta^{18}O$ | Precip | | 42.08 | -123.42 | Ersek et al., 2012 | https://doi.org/10.1038/ncomms2222 |
| Oro Lake | Lake sediment | Diatom | D-assemblage | Precip | | 49.78 | -105.33 | Michels et al., 2007 | https://doi.org/10.1111/j.1365-2486.2007.01367.x |
| Park Range | Lake sediment | Pollen | P-assemblage | Precip | | 40.7 | -106.75 | Parish et al., 2020 | https://doi.org/10.1017/qua.2019.85 |
| Path Lake | Lake sediment | Pollen | P-assemblage | Precip | ±30 | 43.87 | -64.93 | Neil et al., 2014 | https://doi.org/10.1016/j.yqres.2014.01.001 |
| Renner lake | Lake sediment | Sediment | $\delta^{18}O$ | Precip | | 48.78 | -118.19 | Steinman et al., 2012 | https://doi.org/10.1073/pnas.1201083109 |
| Rogers Lake | Lake sediment | Pollen | P-assemblage | Precip | | 41.21 | -72.17 | Marsicek et al., 2013 | https://doi.org/10.1016/j.quascirev.2013.09.001 |
| Sharkey Lake | Lake sediment | Pollen | P-assemblage | Precip | | 44.59 | -93.41 | Shuman et al., 2016 | https://doi.org/10.1016/j.quascirev.2016.03.009 |
| Southern Colorado Plateau | Tree | Tree-rings | Total ring width | Temp, Precip | | 36.56 | -110.12 | Salzer and Kipfmueller 2005 | https://doi.org/10.1007/s10584-005-5922-3 |
| Southern Sierra Nevada | Tree | Tree-rings | Total ring width | Precip | | 37.03 | -119.43 | Touchan et al., 2021 | https://doi.org/10.1007/s00382-020-05548-0 |
| Spruce Pond | Lake sediment | Pollen | P-assemblage | Temp | | 41.24 | -74.2 | Shuman et al., 2016 | https://doi.org/10.1016/j.quascirev.2016.03.009 |
| Steel Lake | Lake sediment | Pollen | P-assemblage | Temp | | 46.97 | -94.68 | Shuman et al., 2016 | https://doi.org/10.1016/j.quascirev.2016.03.009 |
| White Mountains | Tree | Tree-rings | Total ring width | Temp | | 37.45 | -118.17 | Bale et al., 2011 | https://doi.org/10.1016/j.yqres.2011.05.004 |
| Baker lake | Tree | Tree-rings | Total ring width | Temp | | 45.9 | -114.3 | Hughes et al., 2005 | https://doi.org/10.25921/fjr2-x671 |
| Boreal Plateau | Tree | Tree-rings | Total ring width | Temp | | 36.3 | -118.3 | Graumlich et al., 2005 | https://doi.org/10.25921/6asd-pd54 |
| Cirque Peak | Tree | Tree-rings | Total ring width | Temp | | 36.3 | -118.2 | Graybill 1995 | https://doi.org/10.25921/vzv9-nc75 |





**Table A3.** Source and distribution information for precipitation (Precip) and temperature (Temp) across the North American region. (Part-2)

| Region | Material | Fractions | Proxy | Variable | Age uncertainty (years) | Latitude | Longitude | Investigators | Publication |
|---|---|---|---|---|---|---|---|---|---|
| Flint Creek Range | Tree | Tree-rings | Total ring width | Temp | | 46.3 | -113.2 | Hughes et al., 2005 | https://doi.org/10.25921/w5qw-an30 |
| Flower Lake | Tree | Tree-rings | Total ring width | Temp | | 36.5 | -118.2 | Graybill 1995 | https://doi.org/10.25921/n5aj-jg56 |
| French Glacier | Tree | Tree-rings | Total ring width | Temp | | 50.8 | -115.3 | Colenutt et al., 1995 | https://doi.org/10.25921/h234-mz90 |
| Kobuk_Noatak | Tree | Tree-rings | Total ring width | Temp | | 67.1 | -159.6 | King and Graumlich, 2003 | https://doi.org/10.25921/3a2b-f314 |
| Landslide | Tree | Tree-rings | Total ring width | Temp | | 60.2 | -138.5 | Luckman et al., 2006 | https://doi.org/10.25921/jj5b-h521 |
| Mount Washington | Tree | Tree-rings | Total ring width | Temp | | 38.5 | -114.2 | Graybill, 1994 | https://doi.org/10.25921/hy54-1r84 |
| Pearl Peak | Tree | Tree-rings | Total ring width | Temp | | 40.2 | -115.5 | Graybill, 1994 | https://doi.org/10.25921/hy6m-b053 |
| Pintlers | Tree | Tree-rings | Total ring width | Temp | | 46 | -113.4 | Gregory et al., 2011 | https://doi.org/10.1126/science.1201570 |
| Prince William Sound | Tree | Tree-rings | Total ring width | Temp | | 60.5 | -148.3 | Barclay et al., 1999 | https://doi.org/10.1191/095968399672825976 |
| San Franciso Peaks | Tree | Tree-rings | Total ring width | Temp | | 35.3 | -111.4 | Salzer and Kipfmueller, 2005 | https://doi.org/10.1007/s10584-005-5922-3 |
| Sheep Mountain | Tree | Tree-rings | Total ring width | Temp | | 37.2 | -118.1 | Graybill 1995 | https://doi.org/10.25921/q0we-8b37 |
| Siberian Outpost View | Tree | Tree-rings | Total ring width | Temp | | 36.5 | -118.3 | Kipfmueller et al., 2010 | https://doi.org/10.1139/X09-187 |
| Spillway Lake | Tree | Tree-rings | Total ring width | Temp | | 37.8 | -119.2 | King et al., 2000 | https://doi.org/10.25921/v5y4-2w90 |
| Timber Gap Upper | Tree | Tree-rings | Total ring width | Temp | | 36.3 | -118.4 | Graybill 1995 | https://doi.org/10.25921/vzyy-js91 |
| Upper Wright Lakes | Tree | Tree-rings | Total ring width | Temp | | 36.4 | -118.2 | Bunn et al., 2005 | https://doi.org/10.1191/0959683605hl827rp |
| Yellow Mountain Ridge | Tree | Tree-rings | Total ring width | Temp | | 45.3 | -111.3 | King et al., 2000 | https://doi.org/10.25921/vt0z-ah64 |
| Lac Noir | Lake sediment | Pollen | P-assemblage | Temp | | 45.8 | -75.1 | Paquette and Gajewski, 2013 | https://doi.org/10.1016/j.quascirev.2013.06.007 |
| Basin pond | Lake sediment | Pollen | P-assemblage | Temp, Precip | | 44.28 | -70.03 | Gajewski 1988 | https://doi.org/10.1016/0033-5894(88)90034-8 |

**Table A4.** Source and distribution information for AMOC, SST, and ITCZ sensitive tracers.

| Region | Location | Material | Fractions | Proxy | Variable | Age uncertainty (years) | Latitude | Longitude | Investigators | Publication |
|---|---|---|---|---|---|---|---|---|---|---|
| Asia | Klang Cave | Stalagmite | Stalagmite | $\delta^{18}O$ | ITCZ | | 8.33 | 98.73 | Tan et al., 2019 | https://doi.org/10.1073/pnas.1903167116 |
| Asia | Anjohibe Cave | Stalagmite | Stalagmite | $\delta^{18}O$ | ITCZ | | -15.53 | 46.88 | Scroxton et al., 2017 | https://doi.org/10.1016/j.quascirev.2017.03.017 |
| Asia | Clearwater Cave | Stalagmite | Stalagmite | $\delta^{18}O$ | ITCZ | | 4.1 | 114.8333 | Carolin et al. 2016 | https://doi.org/10.1126/science.1233797 |
| Asia | Dongge Cave | Stalagmite | Stalagmite | $\delta^{18}O$ | ITCZ | | 25.283333 | 108.083333 | Wang et al., 2006 | https://doi.org/10.1126/science.1106296 |
| Asia | Heshang Cave | Stalagmite | Stalagmite | $\delta^{18}O$ | ITCZ | | 30.45 | 110.416667 | Hu et al., 2008 | https://doi.org/10.1016/j.epsl.2007.10.015 |
| Asia | Liang cave | Stalagmite | Stalagmite | $\delta^{18}O$ | ITCZ | | -8.533333 | 120.433333 | Scroxton et al., 2022 | https://doi.org/10.1038/s41598-022-21843-8 |
| Asia | Shenqi Cave | Stalagmite | Stalagmite | $\delta^{18}O$ | ITCZ | | 28.93 | 103.1 | Tan et al., 2018 | https://doi.org/10.1016/j.quascirev.2018.07.021 |
| Asia | Tzabnah | Stalagmite | Stalagmite | $\delta^{18}O$ | ITCZ | | 20.733333 | -89.466667 | Elizalde et al., 2010 | https://doi.org/10.1016/j.epsl.2010.08.016 |
| Asia | Wanxiang Cave | Stalagmite | Stalagmite | $\delta^{18}O$ | ITCZ | | 33.316667 | 105 | Zhang et al., 2009 | https://doi.org/10.1126/science.1163965 |
| Asia | Yok Balum Cave | Stalagmite | Stalagmite | $\delta^{18}O$ | ITCZ | | 16.2086 | -89.0735 | Kennett et al., 2012 | https://doi.org/10.1126/science.1226299 |
| Asia | Klang Cave | Stalagmite | Stalagmite | $\delta^{18}O$ | ITCZ | | 8.33 | 98.73 | Chawchai et al., 2021 | https://doi.org/10.1016/j.quascirev.2020.106779 |
| Asia | Hoti Cave | Stalagmite | Stalagmite | $\delta^{18}O$ | ITCZ | | 23.05 | 57.21 | Fleitmann et al., 2022 | https://doi.org/10.1126/science.abg4044 |
| Asia | Botuvera Cave | Stalagmite | Stalagmite | $\delta^{18}O$ | ITCZ | | -27.223333 | -49.155556 | Bernal et al., 2016 | https://doi.org/10.1016/j.epsl.2016.06.008 |
| Asia | Botuvera Cave | Stalagmite | Stalagmite | $\delta^{18}O$ | ITCZ | | -29.2233 | -49.12 | Cruz et al., 2005 | https://doi.org/10.1038/nature03365 |
| Australia | Cave KNI-51 | Stalagmite | Stalagmite | $\delta^{18}O$ | ITCZ | | -15.18 | 128.37 | Denniston et al., 2016 | https://doi.org/10.1038/srep34485 |
| North America | Juxtlahuaca Cave | Stalagmite | Stalagmite | $\delta^{18}O$ | ITCZ | | 17.4 | -99.2 | Lachniet et al., 2012 | https://doi.org/10.1130/G32471.1 |
| North America | Juxtlahuaca Cave | Stalagmite | Stalagmite | $\delta^{18}O$ | ITCZ | | 17.4 | -99.2 | Lachniet et al., 2013 | https://doi.org/10.1073/pnas.1222804110 |
| North America | Juxtlahuaca Cave | Stalagmite | Stalagmite | $\delta^{18}O$ | ITCZ | | 17.44 | -99.16 | Lachniet et al., 2017 | https://doi.org/10.1016/j.quascirev.2016.11.012 |
| Ocean | KNR166-2 11MC-D | Marine sediment | Sediment | $Cd/Ca$ | AMOC | | 24.219667 | -83.295833 | Valley et al., 2022 | https://doi.org/10.1029/2021PA004379 |
| Ocean | MD03-2661 | Marine sediment | Foraminiferal | $\delta^{13}C$ | AMOC | ±40 | 38.89 | -76.4 | Cronin et al., 2010 | https://doi.org/10.1016/j.palaeo.2010.08.009 |
| Ocean | Island of Grimsey | Marine sediment | Molusc | $\delta^{13}C$ | AMOC | ±30-50 | 66.53 | -18.2 | Wanamaker et al., 2012 | https://doi.org/10.1038/ncomms1901 |
| Ocean | KN140-2-51 | Marine sediment | Sediment | Sortable silt | AMOC | ±41.20 | 32.78 | -76.283 | Hoffmann et al., 2018 | https://doi.org/10.1029/2018GL080187 |
| Ocean | NEAP-4K | Marine sediment | Sediment | Sortable silt | AMOC | | 61.4985 | -24.172167 | Thornalley et al., 2013 | https://doi.org/10.5194/cp-9-2073-2013 |
| Ocean | ODP983 | Marine sediment | Sediment | Sortable silt | AMOC | | 62.4035 | -23.640667 | Thornalley et al., 2013 | https://doi.org/10.5194/cp-9-2073-2013 |
| Ocean | Orphan Knoll | Marine sediment | Sediment | Sortable silt | AMOC | | 50.208 | -45.688 | Hoogakker et al., 2011 | https://doi.org/10.1029/2011PA002155 |
| Ocean | MD992251 | Marine sediment | Sediment | Sortable silt | AMOC | | 57.458 | -27.913 | Hoogakker et al., 2011 | https://doi.org/10.1029/2011PA002155 |
| Ocean | GS06-144 08GC | Marine sediment | Sediment | Sortable silt | AMOC | | 60.3197 | -23.966667 | Mjell et al., 2014 | https://doi.org/10.1002/2014PA002737 |
| Ocean | RAPiD-12-1K | Marine sediment | Foraminiferal | $Mg/Ca$ | SST | ±32 | 62.09 | -17.82 | Thornalley et al., 2009 | https://doi.org/10.1038/nature07717 |
| Ocean | RAPiD-17-5P | Marine sediment | Sediment | $Mg/Ca$ | SST | | 61.48 | -19.54 | Sanchez et al., 2014 | https://doi.org/10.1038/ngeo2094 |
| Ocean | RAPiD-35-25B | Marine sediment | Foraminiferal | $Mg/Ca$ | SST | | 38.8863 | -76.3947 | Sanchez et al., 2014 | https://doi.org/10.1002/2013PA002523 |
| Ocean | RD-2209 | Marine sediment | Foraminiferal | $Mg/Ca$ | SST | | 38.89 | -76.4 | Cronin et al., 2010 | https://doi.org/10.1016/j.palaeo.2010.08.009 |
| Ocean | KNR140-2-59GGC | Marine sediment | Foraminiferal | $Mg/Ca$ | SST | | 32.976995 | -76.316001 | Arbuszewski et al., 2010 | https://doi.org/10.1016/j.epsl.2010.10.035 |
| Ocean | ENAM9606 | Marine sediment | Foraminiferal | $Mg/Ca$ | SST | | 55.650334 | -13.985 | Richter et al., 2009 | https://doi.org/10.1016/j.quascirev.2009.04.008 |
| Ocean | Norwegian Sea | Marine sediment | Alkenone | $U^{k'}37$ | SST | | 60.866129 | 3.732566 | Eiriksson et al., 2006 | https://doi.org/10.1177/0959683606hl991rp |
| Ocean | JR51-GC35 | Marine sediment | Sediment | $U^{k'}37$ | SST | | 67.59 | -17.56 | Bendle & Rosell-Melé, 2007 | https://doi.org/10.1177/0959683607073269 |
| Ocean | MD99-2266 | Marine sediment | Foraminiferal | $U^{k'}37$ | SST | ±165 | 66.23 | -24 | Moossen et al., 2015 | https://doi.org/10.1016/j.quascirev.2015.10.013 |





*Author contributions.* Shailendra Pratap conducted the analyses and wrote this manuscript. Yannis Markonis and Cécile Blanchet provided valuable feedback and suggested critical improvements to the work. All authors contributed to the development of scientific ideas and the 470 writing of the article.

*Competing interests.* The authors declare that they have no competing interest.

*Acknowledgements.* Shailendra Pratap (SP) acknowledges support from the Faculty of Environmental Sciences, Czech University of Life Sciences Prague, through the Internal Grant Agency with grants 2021B0006 and 2022B0039. SP also received financial and logistical support from the Helmholtz Visiting Researcher Grant, supported by the Helmholtz Information and Data Science Academy, Germany, with 475 grant number 15650, during his stay at the Helmholtz Centre Potsdam - GFZ German Research Centre for Geosciences. Yannis Markonis was funded by the Czech Science Foundation (Grant 22-33266M), within the project "Investigation of Terrestrial HydrologicAl Cycle Acceleration (ITHACA)". The authors would also like to acknowledge the extensive database provided by NOAA and PANGAEA.





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
