# Peer review of "Changes in precipitation and temperature patterns related to the state of the North Atlantic Ocean during the Medieval Climate Anomaly"

_Climate of the Past, 2024_

## Author Comment (AC1)

**Reply on RC1**

**General comments**

Review of "Changes in precipitation and temperature patterns related to the state of the North Atlantic Ocean during the Medieval Climate Anomaly" (cp-2024-68)

In this paper the authors are interested in temperature and hydroclimate of the MCA and compare a range of proxy data sources with a paleoclimate data assimilation product and a model simulation. In general I think the paper does a decent job of accomplishing the stated goals and the conclusions are in line with the somewhat ambiguous and somewhat conflicting results of the underlying data.

We thank the reviewer for a constructive and positive review. We have addressed the issues raised and provide more detailed responses in the following. In particular, we acknowledge the lack of clarity in some parts of our manuscript and have amended it accordingly.

**Other comments**

1. PaleoView data source: This source appears to just be a way of accessing the TRaCE21ka simulation? Some places in the paper the authors appear to categorize it as a data assimilation product like PHYDA (e.g., lines 270-271) but that is not accurate. They authors need to make it more clear that PaleoView is the TRaCE21ka simulation data alone and is not modified by proxy data in the same way that PHYDA is.

We agree that PaleoView provides access to the TRaCE-21ka transient simulation data and is not a data assimilation product. In the revised manuscript, we have clarified this distinction in the Data and Methods section (lines 160-165) and corrected the relevant statements in the Results and Discussion section. Specifically, we have removed any language that may have implied PaleoView involves data assimilation or is analogous to PHYDA. We now explicitly state that PaleoView represents a purely model-based simulation.

2. Reference to "grid" values throughout the text: I recommend using something like "continental average" (or perhaps some abbreviation indicating regional average) to indicate that this is a regional averaged value. Calling it "grid" is a little strange or unclear what is meant if you are also using "point" estimates because the terms grids and points are often used interchangeably or are at least closely related.

In our study, the term 'grid' was originally used to denote a continental average. However, we have revised the manuscript to clarify the ambiguous use of the term "grid" by employing more precise wording. Specifically, we now state that grid-scale outputs reflect spatially averaged conditions across the entire continent, while point-scale outputs represent values extracted at the exact locations of proxy data collection sites (lines 307-309).

3. Hydroclimate variables: I'm confused as to why the authors used the variable label "P-E" throughout the comparisons between proxy, model, and reconstructed data. Having some proxies be approximately P-E is plausible enough but then the TRaCE21ka data is just temperature or precipitation (so missing E) and PHYDA is PDSI, which is much more complicated than P-E. I think the authors should indicate in Figs 8,9, Table 1 that these values are not actually P-E. This may also account for some of the discrepancies that the authors see in their analyses between the data sources.

We agree that using the term 'P-E' universally could be misleading. To address this concern, we have revised the terminology throughout the manuscript and in the section focusing on comparisons between proxy- and model-derived outcomes, including the labels and captions in Figures 8 and 9 and in Table 1, to clearly indicate the specific variable used in each case. These revisions now reflect the appropriate hydroclimate metric: P-E for the proxy-based records, PDSI for the PHYDA data, and precipitation for the PaleoView simulations. In Table 1, we have adopted the term 'hydroclimate indicator' to refer collectively to these three variables (i.e., P-E, PDSI, and precipitation). The table

caption has also been updated to clarify that this term represents distinct metrics-each serving as a proxy for hydroclimatic state.

4. ITCZ: So the ITCZ is a deep-tropical phenomena yet very little proxy data analyzed here is from the tropics. It was unclear to me how it was justified and used to be an indicator of the ITCZ? Also I noticed that the authors frequently use ITCZ for assessing North Atlantic variability. I don't think this makes sense given that the ITCZ and North Atlantic are dynamically and geographically separate things.

We acknowledge that the ITCZ is inherently a tropical phenomenon. In this study, our objective is to investigate its latitudinal shifts (northward or southward) in response to North Atlantic variability, particularly changes in AMOC and SSTs. We have revised the manuscript to clarify that the ITCZ is used as a response indicator rather than a driver of North Atlantic variability. Given the absence of direct records of past ITCZ shifts, we employ $\delta^{18}O$ values from stalagmites as a proxy for reconstructing ITCZ variability, following the methodology of Tan et al. [2019] and Chawchai et al. [2021]. Our study is the first to compile more than two ITCZ indicators, specifically, 11 $\delta^{18}O$ records from sites in the Northern Hemisphere and 5 from the Southern Hemisphere, all located within the present migration range of the ITCZ across both hemispheres. This spatial framework allows us to estimate hemispheric shifts in the ITCZ and examine their relationship to North Atlantic climate conditions. We have revised the manuscript to clarify this methodological justification and the role of the ITCZ in our study.

**References**

Chawchai S, Tan L, Löwemark L, Wang HC, Yu TL, Chung YC, Mii HS, Liu G, Blaauw M, Gong SY, et al. (2021) Hydroclimate variability of central indo-pacific region during the holocene. Quaternary Science Reviews 253:106779

Tan L, Shen CC, Löwemark L, Chawchai S, Edwards RL, Cai Y, Breitenbach SF, Cheng H, Chou YC, Duerrast H, et al. (2019) Rainfall variations in central indo-pacific over the past 2,700 y. Proceedings of the national academy of sciences 116(35):17201–17206

---

## Author Comment (AC2)

**Reply on RC2**

**General comments**

In this manuscript Pratap and co-authors seek to explore variations in temperature, hydroclimate and ocean circulation, primarily in the north Atlantic during the middle portion of the Common Era. This is achieved through the synthesis of previously-published records and their comparison to a data assimilation product (PHYDA ) and a CCSM3 simulation spanning the past 21,000 years (accessed via PaleoView). The focus on the time from 800-1400 C.E., during the so-called Medieval Climate Anomaly, arguing that it is warm climate.

I am generally supportive of synthesis efforts and data-model comparison efforts. Every source of paleoclimate data has it's strengths and weaknesses, and combining them can often amplify mutual signals and minimize noise. However, the value of such syntheses should be to yield new insight and I find that Pratap and co-authors do not successfully achieve this. While I find no fatal flaws in their analyses, I also do not see that this study brings much new insight to the community. Given that other synthesis studies exist using similar data (e.g Moffa-Sanchez et al., 2019; https://doi.org/10.1029/2018PA003508), the authors do not make a clear case for what this study adds. Many conclusions seem either fairly well established (e.g. warm SST corresponds with warm continental temperature, cool SST drives the ITCZ southward, etc.) or are ambiguous (e.g. "the sensitivity of AMOC tracers across both space and time require further investigation"). Thus, I suggest the authors either more clearly articulate how their work advances understanding relative to previous synthesis studies or hold off on publication until they have a result that does advance knowledge.

We thanks the reviewer for thoughtful evaluation and for acknowledging the robustness of our analyses. While we understand your concern regarding the study's novelty, we would like to highlight several aspects that we believe make our work a valuable contribution to the community. Our study provides a comprehensive centennial-scale comparison of hydroclimate variability during the MCA across both Europe and North America, integrating multiple proxies and model-derived outcomes (i.e., from both assimilated and simulated sources). This dual approach offers a novel spatial perspective, as we examine hydroclimate patterns at both regional and continental scales– a matter of ongoing scientific investigation. By evaluating hydroclimatic coherence across these spatial scales, we offer insights into MCA variability that may inform understanding of present and future hydroclimate variability and oceanic changes under warm climate conditions. Specially, the relationship we observe between temperature and precipitation during a warm climatic phase provides further insight, particularly in the context of ongoing and projected global warming.

Additionally, our findings suggest a possible link between megadrought conditions in North America and southward shifts of the ITCZ, likely driven by low North Atlantic SSTs and weakened AMOC phases. This connection contributes to the limited but growing body of evidence on how ocean-atmosphere interactions influence terrestrial hydroclimate over centennial timescales. We emphasize how North Atlantic variability (i.e., SST and AMOC changes) under warm conditions may affect hydroclimate distributions across tropical and subtropical regions; an aspect that remains underexplored in MCA studies. Finally, our investigation into the combined influence of North Atlantic variability and ITCZ shifts on terrestrial hydroclimate helps disentangle their relative roles. This focus and effort to trace possible teleconnections provide new perspectives on the broader climatic mechanisms shaping hydroclimate variability during a known warm period, with implications for future climate scenarios.

Unlike prior studies such as Moffa-Sánchez et al. (2019), which focus primarily on variability in the northern North Atlantic and Arctic Oceans, our study conducts a coordinated centennial-scale synthesis of hydroclimate patterns across both Europe and North America and examines their links to North Atlantic variability. This cross-continental approach enables comparative insights into spatio-temporal variability. Second, we conducted a model–proxy evaluation, distinguishing between point-scale and grid-scale (continental average) model outputs to better align model resolution with the

spatial distribution of proxy data. This approach reveals how spatial aggregation can mask important local variability. Third, we incorporate a $\delta^{18}$O-based ITCZ reconstruction using records from both hemispheres to examine its relationship with SST and AMOC changes and its influence on regional hydroclimate patterns. Our study is the first to compile more than two ITCZ indicators, specifically, 11 $\delta^{18}$O records from sites in the Northern Hemisphere and 5 from the Southern Hemisphere, all situated within the present migration range of the ITCZ. This hemispheric framework allows us to estimate ITCZ shifts, evaluate their connection to North Atlantic climate variability, and assess their influence on both regional and continental-scale hydroclimate patterns. Fourth, we interpret our findings in the context of warm climate conditions, framing the MCA as a partial analog for current and future warming. For instance, we show that warm periods during the MCA correspond to arid conditions in parts of North America, offering insights into potential hydroclimate responses under modern warming scenarios. Lastly, our study identifies regionally distinct temperature-precipitation associations and highlights spatial mismatches between model outputs and proxies-refinements that may guide future model development and calibration.

To articulate in a more clear way how our work advances understanding relative to previous studies within the revised manuscript, we summarized the above points to the following paragraph that has been introduced at lines 91 to 113 of the introduction.

A more specific concern regards how the authors approach the PHYDA data at its comparison to their work. The authors state that PHYDA is included to "assess the reliability of model-based paleoclimate outputs," and I fear they may be interpreting PHYDA as a model. Consistent with this, they suggest that poor correlations between their data and PHYDA highlight "the need for improvements in model performance." Rather, PHYDA is a data assimilation product that likely includes many of the same datasets considered by Pratap et al, but arguably synthesizes these data in a more sophisticated and physically-realistic way.

We agree with the reviewer's point that PHYDA is a data assimilation product, not a standalone climate model. We appreciate the reminder that it likely includes many of the same datasets as our proxy compilation and that its design reflects a more physically informed synthesis approach. Our previous phrasing may have unintentionally implied that PHYDA is a pure model output or that discrepancies with it reflect deficiencies in model performance. In the revised manuscript, we now clearly describe PHYDA as a paleoclimate reconstruction that integrates proxy data with climate model priors using a data assimilation framework. Clarifications have been made in both the Data and Methods (lines 157–165) and Results (lines 305–325) sections to ensure an accurate representation of PHYDA role and to avoid misinterpretation.